# A dry lunar mantle reservoir for young mare basalts of Chang'e-5

Sen Hu[1 ✉], Huicun He[1], Jianglong Ji[1], Yangting Lin[1 ✉], Hejiu Hui[2,3], Mahesh Anand[4,5], Romain Tartèse[6], Yihong Yan[1], Jialong Hao[1], Ruiying Li[1], Lixin Gu[1], Qian Guo[7], Huaiyu He[7] & Ziyuan Ouyang[8]

The distribution of water in the Moon's interior carries implications for the origin of the Moon[1], the crystallization of the lunar magma ocean[2] and the duration of lunar volcanism[2]. The Chang'e-5 mission returned some of the youngest mare basalt samples reported so far, dated at 2.0 billion years ago (Ga)[3], from the northwestern Procellarum KREEP Terrane, providing a probe into the spatiotemporal evolution of lunar water. Here we report the water abundances and hydrogen isotope compositions of apatite and ilmenite-hosted melt inclusions from the Chang'e-5 basalts. We derive a maximum water abundance of $283 \pm 22 \, \mu g \, g^{-1}$ and a deuterium/hydrogen ratio of $(1.06 \pm 0.25) \times 10^{-4}$ for the parent magma. Accounting for low-degree partial melting of the depleted mantle followed by extensive magma fractional crystallization[4], we estimate a maximum mantle water abundance of $1–5 \, \mu g \, g^{-1}$, suggesting that the Moon's youngest volcanism was not driven by abundant water in its mantle source. Such a modest water content for the Chang'e-5 basalt mantle source region is at the low end of the range estimated from mare basalts that erupted from around 4.0 Ga to 2.8 Ga (refs. [5,6]), suggesting that the mantle source of the Chang'e-5 basalts had become dehydrated by 2.0 Ga through previous melt extraction from the Procellarum KREEP Terrane mantle during prolonged volcanic activity.

Water abundance in the lunar mantle places strict constraints on high-temperature processes, including the Moon-forming giant impact[1], the ensuing crystallization of the lunar magma ocean[7] and the longevity of volcanism on the Moon[2]. On the basis of analyses carried out since the Apollo missions, the Moon was long thought to be anhydrous. Advances in in situ analytical techniques over the past decade have allowed the analysis of water abundances at the microscale in various lunar samples, including in olivine-hosted and pyroxene-hosted melt inclusions in mare basalts[8–12], apatite in mare basalts and highlands samples[13–20], pyroclastic glass beads[21,22] and anorthosites[23,24]. The estimates of water abundances for the mantle source regions of these samples span a wide range, from about $0.3 \, \mu g \, g^{-1}$ to $200 \, \mu g \, g^{-1}$ (ref. [25]), which suggests that the lunar interior is not as anhydrous as once thought. Although the large variations in the water abundance estimates for lunar mantle sources could partially be due to the assumptions involved in these calculations, many questions remain regarding the origin(s) and distribution of water in the Moon's interior[25,26]. Variations in estimated mantle water abundances may be indicative of geographic and/or temporal diversity as these samples were collected from different regions and crystallized between around 4.0 billion years ago (Ga) to 2.8 Ga (refs. [5,6]). Hence, studying additional sample collections of younger basalts from different regions can provide critical additional constraints on the spatiotemporal evolution of water

in the Moon. The large range of mantle water abundance estimates could also be affected by the mixing of endogenic water with various exogenic water sources, that is, asteroids, comets and solar wind[19,26,27], and/or by the interplay between many processes, including volatile degassing, partial melting, fractional crystallization, impacting, mixing with potassium (K), rare-earth elements (REE) and phosphorus (P) (KREEP)-rich components, and spallation[11,15,25,27–29]. It is thus crucial to combine in situ analysis of water abundances and hydrogen isotope compositions with detailed contextual petrographic information.

The Chang'e-5 (CE5) mission successfully returned 1.731 kg of lunar soil samples from young mare basalt units dated at 2.0–1.2 Ga using crater-counting chronology[30,31]. These CE5 samples have now been precisely dated at $2,030 \pm 4$ million years ago (Ma) using the lead (Pb)–Pb isotope isochron technique[3]. The CE5 basalts are thus much younger than the youngest lunar basalt samples dated so far (2.8 Ga (ref. [5])). The young basalt unit is located in northwestern Oceanus Procellarum, at the northwestern edge of the Procellarum KREEP Terrane (PKT), which is far from all landing sites of the Apollo and Luna missions (Extended Data Fig. 1). The PKT region is also thought to have enhanced concentrations of two major radioactive heat-producing elements, uranium (U) and thorium (Th), and other incompatible elements[32]. Water behaves as a typical incompatible element during magmatic processes[33] and thus is expected to be enriched in the PKT as well. Hence, the CE5 basalts

[1]Key Laboratory of the Earth and Planetary Physics, Chinese Academy of Sciences, Beijing, China. [2]State Key Laboratory for Mineral Deposits Research & Lunar and Planetary Science Institute, School of the Earth Sciences and Engineering, Nanjing University, Nanjing, China. [3]CAS Center for Excellence in Comparative Planetology, Hefei, China. [4]School of Physical Sciences, The Open University, Milton Keynes, UK. [5]Department of Earth Sciences, The Natural History Museum, London, UK. [6]Department of Earth and Environmental Sciences, The University of Manchester, Manchester, UK. [7]State Key Laboratory of Lithospheric Evolution, Chinese Academy of Sciences, Beijing, China. [8]Center for Lunar and Planetary Sciences, Institute of Geochemistry, Chinese Academy of Sciences, Guiyang, China. ✉e-mail: husen@mail.iggcas.ac.cn; linyt@mail.iggcas.ac.cn

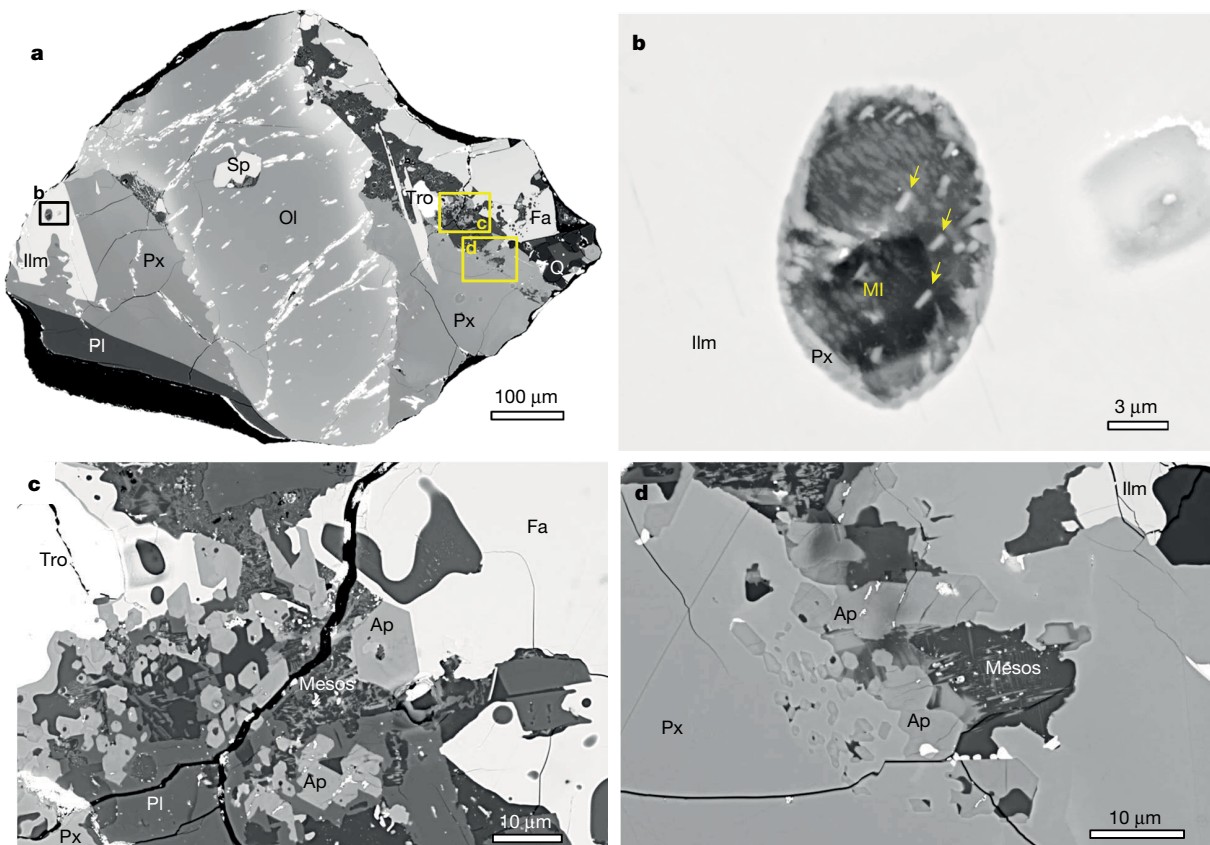

**Fig. 1 | Backscattered electron images of ilmenite-hosted melt inclusions and apatite from a CE5 basalt clast. a**, The basalt clast (406-010,023) embedded in the metal mount is mainly composed of olivine (Ol), pyroxene (Px), plagioclase (Pl) and ilmenite (Ilm), with minor fayalite (Fa), troilite (Tro), spinel (Sp), apatite (Ap) and silica (Q). The locations of **b**, **c** and **d** are outlined by the rectangles in **a**. **b**, High-resolution BSE image of a melt inclusion hosted in ilmenite. This melt inclusion shows a post-entrapment crystallization texture with occurrences of submicrometre-sized pyroxene and merrillite (yellow arrows). **c**, High-resolution BSE image of apatite in the interstitial areas. Many euhedral apatite grains, up to 10 μm in length, coexist with fine-grained plagioclase, fayalite and mesostasis (Mesos). **d**, Small apatite grains also occur at the rims of pyroxene, coexisting with mesostasis. The bright pits in cracks are the remains of the gold coating.

provide a unique opportunity to constrain the water inventory of a newly sampled region of the Moon's interior, and thus yields crucial information to account for the prolonged activity of lunar magmatism.

## Water in CE5 apatite and melt inclusions

We studied a total of 23 basalt clasts (0.2–1.5 mm in size) from two CE5 soil samples (CE5C0100YJFM00103, about 1 g; CE5C0400YJFM00406, about 2 g; Extended Data Table 1). The basalt clasts show variable textures, including subophitic, poikilitic and equigranular, similar to those observed for other basalt clasts in CE5 soil samples[3,4], and are mainly composed of pyroxene and plagioclase with less abundant olivine and ilmenite (Fig. 1, Supplementary Figs. 1, 2). These basalt clasts are probably representative of various portions of the same lava flow, based on their identical mineral chemistry and geochemistry[4] and their well defined Pb–Pb isochron[3]. The textures of ilmenite in the clasts indicate that this phase began to crystallize early from the melt and continued until the last stages of melt evolution (Supplementary Fig. 1). Ilmenite-hosted melt inclusions are 4–50 μm in diameter and show post-entrapment crystallization textures (0–52%) (Fig. 1, Extended Data Table 3, Supplementary Fig. 1). Apatite occurs as euhedral grains (mostly less than 10 μm) mainly in the fine-grained interstitial materials, with a few euhedral crystals enclosed in the margins of pyroxene (Fig. 1) and iron(II) oxide (FeO)-rich olivine (Supplementary Fig. 2). Apatite is the main hydroxyl (OH)-bearing phase, fluorine (F) rich and chlorine (Cl) poor, similar to those from Apollo mare basalts (Supplementary

Fig. 6), and is an accessory phase in these basalt clasts comprising approximately 0.4 vol% modal abundance (Supplementary Table 1). Eight ilmenite-hosted melt inclusions and several apatite grains were located and selected for in situ analysis (Fig. 1, Extended Data Table 1, Supplementary Figs. 1, 2). The water abundances and hydrogen isotope compositions of the ilmenite-hosted melt inclusions, apatite and nominally anhydrous clinopyroxene were measured using a nanoscale secondary ion mass spectrometer (NanoSIMS) instrument (Methods).

The majority of apatite grains contain water abundances ranging from $555 \pm 31$ μg g$^{-1}$ to $4{,}856 \pm 217$ μg g$^{-1}$ (average $1{,}921 \pm 910$ μg g$^{-1}$, $1\sigma$) with δD values ranging from $275 \pm 85$‰ to $1{,}022 \pm 87$‰ (average $578 \pm 208$‰, $1\sigma$) (δD = $1{,}000 \times$ ([D/H$_{sample}$]/[D/H$_{standard}$] − 1, where D is deuterium and H is hydrogen), using Vienna standard mean ocean water as the standard) (Fig. 2, Extended Data Table 2), which overlap with apatite water abundances and δD values measured in high-titanium (Ti) and low-Ti Apollo basalts[11,15–17,28,29,34–37]. Three apatite analyses yielded lower water abundances ($110 \pm 13$ μg g$^{-1}$ to $235 \pm 19$ μg g$^{-1}$; Extended Data Table 2), with corresponding δD values indistinguishable from the majority of other analyses. As apatite is the major water-bearing phase in the CE5 basalts, a water abundance of $7 \pm 3$ μg g$^{-1}$ for the bulk composition of the CE5 basalts was calculated from the average water content of apatite and its modal abundance of approximately 0.4 vol% (Methods). It is noted that this water abundance is not the original water abundance in the CE5 basaltic magma before eruption, but represents the residual water abundance after magma degassing at the time of apatite crystallization[14]. Furthermore, the apatite δD values

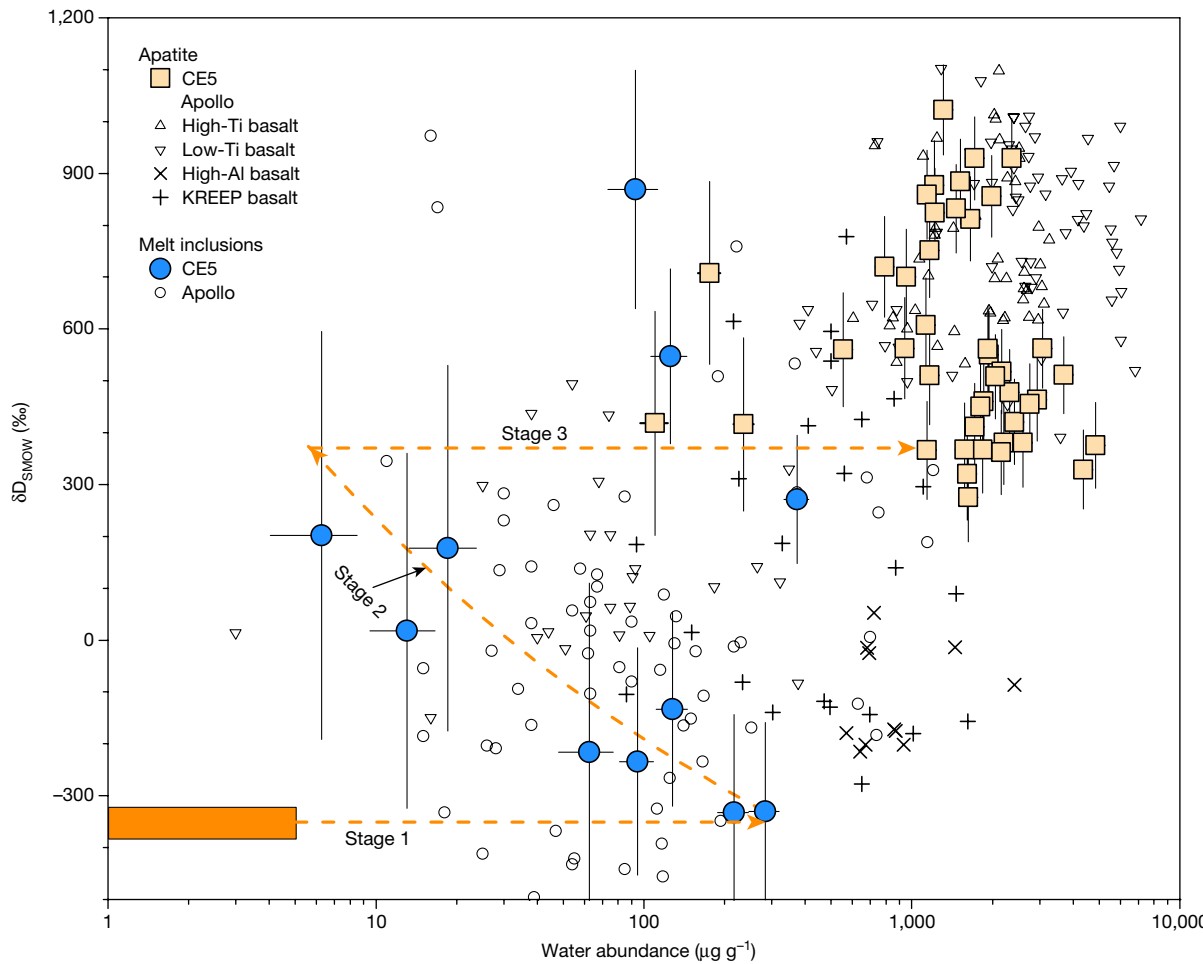

**Fig. 2 | Water abundance and δD of apatite and ilmenite-hosted melt inclusions from CE5 basalts.** The majority of melt inclusions show a negative correlation between the water abundances and δD values, except for three melt inclusions with higher δD values plotting close to the range of apatite. The dotted lines indicate a three-stage evolutionary path, starting with 2–3% partial melting of the mantle source of CE5 basalts, followed by 43–88% fractional crystallization (stage 1), H₂ degassing from the basaltic melts accompanied by deuterium enrichment (stage 2) and crystallization of apatite from the residual melts, possibly accompanied by further H₂ degassing (stage 3). Apatite and melt inclusion (olivine hosted or pyroxene hosted) data from Apollo samples (Supplementary Table 5) are shown for comparison. The CE5 data have been corrected for a nominal CRE of 50 Ma (Methods). The error bars are 2σ.

reflect the signature of the last residual melt after precipitation of most constituent minerals, and the observed large deuterium-enrichment is probably the result of degassing of hydrogen-bearing species from the melt, mostly in the form of H₂ under the reducing conditions that are typical for lunar volcanic products[15].

The ilmenite-hosted melt inclusions contain water abundances of 6 ± 2 μg g⁻¹ to 370 ± 21 μg g⁻¹ after correction of post-entrapment crystallization, and show a wide range of δD values from −330 ± 190‰ to 869 ± 230‰ after correcting for the effects of cosmic ray spallation (Fig. 2, Extended Data Table 3). Cosmic-ray spallation mainly produces deuterium, and can have a large effect on δD values, especially for water-poor (less than 30 μg g⁻¹) melt inclusions[38]. Cosmic-ray exposure (CRE) ages determined for various Apollo samples are mostly younger than around 200 Ma (ref. [39]), but have not yet been measured for CE5 samples. We have modelled the spallation effects on δD values of the melt inclusions, using CRE ages of 10 Ma, 50 Ma, 100 Ma and 200 Ma (Extended Data Fig. 4, Supplementary Table 4). Using CRE ages of 100 Ma and 200 Ma yields noticeable over-correction of δD values as the resulting values are even more deuterium depleted than the currently accepted hydrogen isotope composition of the lunar mantle (Extended Data Fig. 4). As determined from Apollo samples, lunar regolith from a depth of about 9 mm is thought to overturn at least once in about 10 Myr (ref. [40]),

which suggests that it is reasonable to assume a CRE age of around 50 Ma for CE5 samples. With a 50-Ma CRE age correction, the melt inclusions with the lowest water abundances yield corrected δD values of 200 ± 390‰ that overlap with the lowest δD value measured for apatite. Importantly, this correction does not greatly affect the δD values of water-rich melt inclusions nor those of apatite grains (Extended Data Fig. 4, Supplementary Table 4). Moreover, spallation by cosmic rays has little effect on water abundances (Extended Data Fig. 4, Supplementary Table 4). After correction for spallation, the melt inclusions with δD ≤ 200‰ show a negative correlation between water abundances (6 ± 2 μg g⁻¹ to 283 ± 22 μg g⁻¹) and δD values (−330 ± 190‰ to 200 ± 390‰). In addition, three melt inclusion analyses with higher δD values (271 ± 124‰ to 869 ± 230‰) overlap with the data for the water-poor apatite grains (Fig. 2, Extended Data Table 3). These observations provide convincing evidence that ilmenite-hosted melt inclusions with δD ≤ 200‰ have recorded the progressive evolution of melts undergoing H₂ degassing, resulting in considerable deuterium enrichment during crystallization of the CE5 basalts[15,41]. Hydrogen diffusion out of melt inclusions is another process by which D/H ratios can be fractionated as reported for melt inclusions in olivine and pyroxene from Apollo basalts[11]. At present, there is no constraint on the diffusion rate of water in ilmenite-hosted melt inclusions. The lowest δD value of

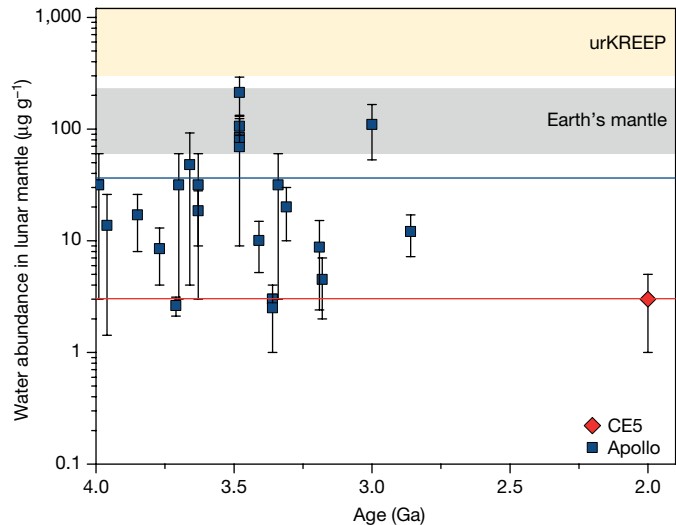

**Fig. 3 | Variation of lunar mantle water abundance estimates with time.** The maximum mantle water abundance at 2.03 Ga, which was estimated in this study using CE5 basalts, plots at the lower end of mantle water abundance estimates for Apollo samples and lunar meteorites formed between around 4.0 Ga and 2.8 Ga. All data are plotted as average values with the error bars representing the ranges of estimates. Estimates for the water abundances in the last dregs of the lunar magma ocean (urKREEP)[25] and Earth's primitive mantle[50] are shown for comparison. The red and dark blue solid lines represent average values estimated from CE5 basalts and previous lunar samples, respectively. It is noted that the vertical axis is a log scale. Literature data are provided in Extended Data Table 5.

approximately −330‰ measured in ilmenite-hosted melt inclusions suggests that the D/H ratios have been well preserved in the melt inclusions without significant exchange of hydrogen isotopes with the deuterium-enriched residual melt through diffusion.

## Magmatic history and source

The lowest $\delta D$ values (−330 ± 190‰) of the melt inclusions are within the ranges of $\delta D$ values for most types of chondrite and the lunar mantle $\delta D$ (0 ± 200‰) estimated from analyses of various lunar samples[11,16,18–20,22,24]. This similarity suggests that the melt with the lowest $\delta D$ was trapped in the early stages of magma crystallization before substantial degassing of water in the form of $H_2$ (refs. [11,15,41]). In contrast, the melt inclusions with higher $\delta D$ (more than 270‰) contain substantial water and overlap with the water-poor apatite values (Fig. 3). This observation can be explained by late crystallization of these ilmenite grains, when deuterium-enriched water was concentrated in the residual melt and apatite started crystallization.

The most deuterium-depleted melt inclusions, as discussed, probably captured signatures of the parent magma before notable loss of water by degassing in the form of $H_2$. Hence, the high water abundance (283 ± 22 µg g$^{-1}$; Extended Data Table 3) of these melt inclusions can be regarded as the maximum water content of the parent magma. Alternatively, the water abundance of the parent magma can also be estimated from a water content of 7 ± 3 µg g$^{-1}$ for the bulk CE5 basalts through calibration for degassing loss of 98–99% water in the form of $H_2$ based on the accompanying $\delta D$ increasing from −330‰ to the average apatite $\delta D$ of 578 ± 208‰ ($1\sigma$) (Extended Data Table 2, Methods). This calculation yields an estimate for the water abundance of the parent magma of 600 ± 400 µg g$^{-1}$, which is consistent within errors with that based on the deuterium-depleted melt inclusions. We thus use the better constrained water abundance from the deuterium-depleted melt inclusions (283 ± 22 µg g$^{-1}$; Extended Data Table 3) as the maximum water abundance of the parent basaltic magma.

The CE5 parental magma was derived from a depleted lunar mantle source not associated with a KREEP component, based on its low initial $\mu$ value ($^{238}U/^{204}Pb$ ratio; 680 ± 20)[3], low initial $^{87}Sr/^{76}Sr$ ratio (0.69934 to 0.69986) and high positive $\varepsilon_{Nd}(t)$ value (7.9 to 9.3)[4]. $\varepsilon_{Nd}(t) = ((^{143}Nd/^{144}Nd)_{sample(t)}/(^{143}Nd/^{144}Nd)_{CHUR} − 1) \times 10,000$, where $(^{143}Nd/^{144}Nd)_{sample(t)}$ and $(^{143}Nd/^{144}Nd)_{CHUR}$ are the Nd isotopic compositions of the sample at its formation time ($t$) and the chondritic uniform reservoir, respectively. The highly elevated abundances of REE and Th, and high FeO and moderate titanium dioxide ($TiO_2$) concentrations of the CE5 parent magma match a model of low-degree (2–3%) partial melting followed by moderate-to-extensive (43–88%) fractional crystallization[4]. Accordingly, the maximum water concentration in the lunar mantle source beneath the CE5 landing site can be estimated at 1–5 µg g$^{-1}$, corresponding to a maximum water abundance of approximately 280 µg g$^{-1}$ in the derived parent magma.

Our analyses of apatite and melt inclusions allow the evolution of CE5 basalts to be divided into three stages. In stage 1, the mantle source region underneath the CE5 landing site with 1–5 µg g$^{-1}$ water experienced low-degree (2–3%) partial melting followed by moderate-to-extensive-degree (43–88%) fractional crystallization[4], thereby generating a parent magma with approximately 280 µg g$^{-1}$ water and preserving its mantle-derived $\delta D$ of −330‰. This maximum water abundance recorded in the melt inclusions captured by the earliest-formed ilmenite analysed here yields our best estimate for the water abundance of the parent magma. In stage 2, $H_2$ degassing from the parent magma occurred during its ascent to shallower depths and during eruption, and was accompanied by the crystallization of ilmenite that entrapped melts at various stages of magmatic evolution. Extensive $H_2$ degassing[15,40] could have occurred in the reduced lunar environment[42,43], resulting in appreciable D/H fractionation from −330‰ up to ~300‰. In stage 3, apatite crystallized from the residual melts that became enriched in water, halogens and other incompatible species, after most nominally anhydrous silicates and ilmenite had formed.

## Evolution of lunar mantle water

The maximum water abundance of 1–5 µg g$^{-1}$ estimated for the mantle source of CE5 basalts is notably at the lower end of the mantle water abundances derived from Apollo basalts and lunar meteorites[9–11,15,25] (Fig. 3). These new indications of a water-poor lunar mantle at 2 Ga carry important implications for understanding late volcanism on the Moon. Such a water-poor mantle source for CE5 basalts excludes the possibility that a high water abundance in the lunar mantle reservoir, by lowering its melting point, could be one of the main causes of the prolonged volcanic activity in this part of the PKT.

Our observations indicate that the water abundance in the Moon's interior may have to some extent decreased from 4.0–2.8 Ga to 2.0 Ga (Fig. 3). Such a systematic loss of water over time could be the result of prolonged magmatic activity in the PKT, where multiple water-bearing melt-extraction episodes from the PKT mantle reservoir occurred but did not fractionate D/H significantly[44]. In the northwestern PKT region, in close proximity to the CE5 landing site, up to ten basaltic units ranging in crater-counting age from 3.7 Ga to 1.2 Ga have been identified[45], although it is difficult to be certain that all these units were derived from the same mantle source region. Nevertheless, such a dehydration partial melting process has also been observed in Earth's mantle[46,47].

Alternatively, the wide range of the water abundance estimates for the mantle source regions of all studied lunar basalts may reflect a heterogeneous distribution of water in the Moon's interior. Furthermore, the water abundance estimates could be affected by possible contamination of some volcanic products by KREEP components during either magma transport or in their mantle source regions during convective overturn of the lunar magma ocean[48,49]. However, it has been demonstrated that CE5 basalts were not contaminated by KREEP components[4].

Thus, our estimate of the mantle water abundance based on CE5 basalts in the PKT region provides a critical spatiotemporal constraint on the distribution of water in the lunar interior. Nonetheless, it remains an enigma to explain the basaltic volcanism sustained as late as 2.0 Ga at the CE5 landing site, the mantle source of which is depleted in the heat-producing elements, U, Th and K, relative to the bulk silicate Moon[4] and is water poor.

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

## Methods

### Sample preparation

Two CE5 lunar soils (CE5C0100YJFM00103, about 1 g; CE5C0400YJFM00406, about 2 g) allocated by the China National Space Administration were used in this study. Both samples were scooped by the robotic arm of the CE5 lander and separated into different packages in the ultraclean room at the extraterrestrial sample curation centre of the National Astronomical Observatories, Chinese Academy of Sciences. Approximately 240 soil fragments with grain sizes varying from about 100 μm to about 1 mm were sieved and hand-picked under a binocular microscope in the ultraclean room at the Institute of Geology and Geophysics, Chinese Academy of Sciences (IGGCAS). Then, about two-thirds of the picked grains were prepared as eight tin–bismuth metal–alloy mounts following the method of Zhang et al.[51] and the other third was mounted in epoxy and prepared into three polished thin sections (about 100 μm in thickness). The polished metal mounts and thin sections were cleaned using ultrapure water and anhydrous ethanol and then dried at 70 °C in a baking oven overnight. The details of apatite and ilmenite-hosted melt inclusions from 23 CE5 basalt clasts and fragments are summarized in Extended Data Table 1.

### Scanning electron microscope observation

Petrographic observations and elemental mapping were carried out using field emission scanning electron microscopes (FE-SEMs) using FEI Nova NanoSEM 450 and Thermofisher Apreo instruments at the IGGCAS, using electron beam currents of 2 nA to 3.2 nA and an acceleration voltage of 15 kV. Energy dispersive spectroscopy X-ray maps were collected for each basaltic clast to locate P-bearing phases. The phosphates were then observed at higher magnification in back-scattered electron (BSE) images. The modal abundance of apatite from various CE5 basalt clasts were counted by the exposed surface areas (Supplementary Table 1). The prepared sections were initially coated with gold to identify apatite and melt inclusions for in situ NanoSIMS measurement of the water content and hydrogen isotopes. After NanoSIMS measurement, the samples were re-coated with carbon and observed by SEM to confirm the positions of the NanoSIMS spots.

### Electron probe microanalysis

After the NanoSIMS analyses, we used a JEOL JXA-8100 electron probe micro-analyser (EPMA) at the IGGCAS to quantify the major and minor elemental abundances in phosphates, melt inclusions in ilmenite and associated mafic minerals (that is, clinopyroxene, olivine, plagioclase and ilmenite). The samples were coated with carbon. The operating acceleration voltage was 15 kV and the beam current was 20 nA. The EPMA analyses were carried out after the NanoSIMS measurements to avoid possible H loss due to bombardment by the electron beam[35]. The EPMA standards were natural albite (sodium (Na) and aluminium (Al)), bustamite (manganese (Mn)), diopside (calcium (Ca), silicon (Si) and magnesium (Mg)), apatite (P), K-feldspar (K), tugtupite (Cl), synthetic fluorite (F), rutile (Ti), iron(III) oxide ($Fe_2O_3$; Fe), vanadium pentoxide ($V_2O_5$; V), nickel oxide (NiO; Ni) and chromium(III) oxide ($Cr_2O_3$; Cr). Na, K, F and Cl were first measured to minimize possible loss of volatiles by electron beam irradiation. The detection limits were ($1\sigma$) 0.01 wt% for Cl and sulfur (S), 0.02 wt% for Na, Mg, Al, Cr, K, Si, Mn, Ca and Fe, 0.03 wt% for F, barium (Ba), Ni and Ti, and 0.04 wt% for P. A program based on the ZAF (Z, atomic number; A, absorption of X-rays in the specimen; F, fluorescence caused by other X-rays generated in the specimen) procedure was used for data correction. The EPMA data obtained for apatite, melt inclusions in ilmenite and the coexisting silicates are listed in Supplementary Table 2.

### In situ water abundance and hydrogen isotope analysis

**Apatite and melt inclusions.** The hydrogen isotopes and water contents of apatite and melt inclusions enclosed in ilmenite from the CE5

basaltic clasts were measured with a CAMECA NanoSIMS 50L at IGGCAS. The samples were coated with gold, loaded in sample holders together with the standards and baked overnight at about 60 °C in the NanoSIMS airlock. The holders were then stored in the NanoSIMS sample chamber to improve the vacuum quality and minimize the H background[52–54]. The vacuum pressure in the analysis chamber was $2.8 \times 10^{-10}$ torr to $3.0 \times 10^{-10}$ torr during analysis. Each 15 μm × 15 μm analysis area was pre-sputtered for 2 min with a $Cs^+$ ion beam current of 2 nA to remove the surface coating and potential contamination. During analysis, the secondary anions $^1H^-$, $^2D^-$ and $^{12}C^-$ were simultaneously counted by electron multipliers and $^{16}O^-$ by a Faraday cup from the central 3 μm × 3 μm areas using the NanoSIMS blanking technique. A 44-ns dead time was corrected for all electron multipliers, while the electron multiplier noise (<$10^{-2}$ counts per second) was ignored. We used a primary ion beam current of about 0.5 nA for analysis, corresponding to a beam size of about 500 nm in diameter. The charging effect on the sample surface was compensated for by an electron gun during analysis.

A chip of the anhydrous San Carlos olivine reference with a reported water content of 1.4 μg g$^{-1}$ (ref. [55]) was used for H background (bg) corrections, following the relationship: $H/O_{bg} = (H_{counts} - H_{bg})/O_{counts}$ and $D/H_{measured} = (1 - f) \times D/H_{true} + f \times D/H_{bg}$, where O is oxygen and $f$ is the proportion of H emitted from the instrumental background[56]. Here $D/H_{bg}$ was 3.36 ($\pm 0.55$) $\times 10^{-4}$ and $H_{bg} = 689 \pm 139$ counts per second (2 s.d., $N = 11$, corresponding to a water background abundance of $25 \pm 8$ μg g$^{-1}$ (2 s.d.)). After background subtraction, the water abundances of apatite grains and melt inclusions were calculated from the background-subtracted H/O ratios multiplied by the slope of the calibration line (Extended Data Fig. 2), which were determined by measuring two apatite standards, Durango apatite (water ($H_2O$) = 0.0478 wt% and $\delta D = -120 \pm 5‰$)[29,57] and Kovdor apatite ($H_2O = 0.98 \pm 0.07$ wt% and $\delta D = -66 \pm 21‰$)[58], the southwest Indian French transect mid-ocean ridge basalt (SWIFT MORB) glass ($H_2O = 0.258$ wt% and $\delta D = -73 \pm 2‰$), and two basaltic glasses, 519-4-1 ($H_2O = 0.17$ wt%)[52] and 1833-11 ($H_2O = 1.2$ wt%)[52] (Supplementary Table 3). Corrections for instrumental mass fractionation (IMF) on hydrogen isotopic compositions of both apatite and melt inclusions were conducted using the Kovdor apatite standard, and monitored by analysing both the Durango apatite and SWIFT MORB glass standards during the whole analytical session (Extended Data Fig. 3). The matrix effects on water abundance and IMF on hydrogen isotopic composition are the same between apatite and silicate glass within analytical uncertainties[54]. Hydrogen isotopic compositions are given using the delta notation, $\delta D = ((D/H)_{sample}/(D/H)_{SMOW}) - 1) \times 1,000$, where SMOW is the standard mean ocean water with a D/H ratio of $1.5576 \times 10^{-4}$. More technical details can be found in Hu et al.[53,54]. All data are reported with their $2\sigma$ uncertainties that include the reproducibility of the D/H measurements on the reference materials, the uncertainty of the $H_2O$ background subtraction and the internal precision on each analysis (Extended Data Tables 2, 3, Supplementary Table 4). The raw measured D/H ratios were corrected for the background, followed by correction for IMF.

**Clinopyroxene.** The water abundance of clinopyroxene from the CE5 basaltic clasts was measured with the CAMECA NanoSIMS 50L using an identical instrument setup to that described above. We used a higher $Cs^+$ primary beam current of 7 nA to improve the $^1H^-$ counts on clinopyroxene and reduce the background. Each 25 μm × 25 μm analysis area was pre-sputtered for about 2 min with the same analytical beam current to remove the surface coating and potential contaminations. The secondary ion signals from the central 7 μm × 7 μm areas were counted with 50% blanking of outermost regions. San Carlos olivine ($H_2O = 1.4$ μg g$^{-1}$ (ref. [55])) was used to determine the instrumental background of $H_2O$, which was $5 \pm 2$ μg g$^{-1}$, about five times lower than the $H_2O$ background during apatite and melt inclusions analyses ($25 \pm 8$ μg g$^{-1}$). The analytical results are listed in Extended Data Table 4.

**Correction of water abundances and D/H ratios for spallation effects.** The measured D/H ratios have been corrected for the potential effects of cosmic-ray spallation, using a deuterium production rate of $2.17 \times 10^{-12}$ mol D g$^{-1}$ Ma$^{-1}$ (ref. [59]) for melt inclusions and $9.20 \times 10^{-13}$ mol D g$^{-1}$ Ma$^{-1}$ (ref. [60]) for apatite. The correction errors induced by deuterium spallation are around 50% on δD and negligible on water content[21]. The CRE ages determined for most Apollo samples are younger than about 200 Ma (ref. [39]), although some ages are up to 400 Ma (ref. [56]). As no CRE age is yet available for the CE5 basaltic clasts, we modelled the effects of corrections for CRE ages of 10 Ma, 50 Ma, 100 Ma and 200 Ma (Extended Data Table 3, Extended Data Fig. 4, Supplementary Table 4). The corrected δD values for the melt inclusions with low water abundances appear to be overcorrected for CRE ages of 100 Ma and 200 Ma, as indicated by unusually low δD values. The corrected δD values for the low-water (less than 50 µg g$^{-1}$) melt inclusions, using a CRE age of 50 Ma, overlap with the lowest δD values of apatite (approximately 300‰; Fig. 2). This observation is consistent with the late capture of the low-water melt inclusions by ilmenite followed by the crystallization of the high-deuterium apatite. The uncertainty in the CRE age mainly affects the negative correlation between δD values and water abundances of the low-deuterium melt inclusions, with only minor effects on estimating the maximum water abundances for the CE5 parent magma and the mantle source.

**Degassing modelling.** The hydrogen isotope fractionation during volatile loss into a vacuum is given by $\alpha^2 = M1/M2$, where M1 and M2 are the masses of the volatile phase isotopologues. The change of the isotopic composition of H during volatile loss by Rayleigh fractionation is given by $R = R_0 \times f^{(\alpha-1)}$, where $R_0$ and $R$ are the initial and final D/H ratios for a fraction $f$ of remaining hydrogen[41]. Degassing of $H_2$ (M1 = 2 for $H_2$ and M2 = 3 for HD) yields an $\alpha$ value of about 0.8165, and degassing of $H_2O$ (M1 = 18 for $H_2O$ and M2 = 19 for HDO) yields an $\alpha$ value of about 0.9733 (ref. [41]; Extended Data Fig. 4).

**Petrography and mineral chemistry of CE5 basalts**
**Petrography.** Approximately 40% of the lithic clasts on a total of 13 sample mounts used in this study are basalt clasts, consisting mainly of pyroxene, plagioclase, olivine and ilmenite, with minor silica, troilite, Si–K-rich mesostasis, apatite and trace merrillite (Supplementary Figs. 1, 2). The basalt clasts display subophitic, poikilitic, porphyritic and equigranular textures (Supplementary Figs. 1, 2), similar to those reported by ref. [4]. Most pyroxene gains are compositionally zoned with dark and low FeO cores and bright and high FeO rims in BSE images (Supplementary Fig. 1) and energy dispersive spectroscopy. Ilmenite grains occur as laths partially enclosed by pyroxene, suggestive of early crystallization (Supplementary Fig. 1). Eight melt inclusions in ilmenite were identified with a diameter of about 4 µm to about 50 µm and circular shapes (Extended Data Table 1, Supplementary Fig. 1). Some of the melt inclusions experienced partial post-entrapment crystallisation (0–52%) with pyroxene and merrillite embedded in a glassy matrix (Supplementary Fig. 1). Most grains of apatite occur in the fine-grained interstitial materials, coexisting with fayalite and K–Si-rich mesostasis (Supplementary Fig. 2). A few euhedral grains of apatite are enclosed in the margins of pyroxene and FeO-rich olivine (Supplementary Fig. 2). Most grains of apatite are smaller than 10 µm, and the detailed information is summarized in Extended Data Table 1.

**Mineral chemistry.** Both pyroxene and olivine from various CE5 basalt clasts are chemically zoned, with higher FeO contents at the rims (Fs about 85.9 mol% and Fa about 98.6 mol%) than in the cores (En about 39.6 mol% and Fa about 43.0 mol%), where Fs = 100 × Fe/(Fe + Ca + Mg) mol%, En = 100 × Mg/(Fe + Ca + Mg) mol%, and Fa = 100 × Fe/(Fe + Mg) mol% (Supplementary Figs. 3, 4, Supplementary Table 1). Plagioclase is relatively homogeneous with a composition of $An_{74.8-92.3}Ab_{7.4-21.6}Or_{0.3-4.7}$,

where An = 100 × Ca/(Ca + Na + K) mol%, Ab = 100 × Na/(Ca + Na + K) mol%, and Or = 100 × K/(Ca + Na + K) mol% (Supplementary Fig. 4, Supplementary Table 1). Ilmenite has a homogeneous composition of 52.9 wt% $TiO_2$ and 44.9 wt% FeO, with minor silicon dioxide ($SiO_2$; <0.45 wt%), $Cr_2O_3$ (<0.32 wt%), manganese oxide (MnO; 0.34–0.47 wt%) and magnesium oxide (MgO; <1.47 wt%) (Supplementary Table 2). Most of the melt inclusions in ilmenite are $SiO_2$ rich (61.2–77.1 wt%), and have wide ranges of aluminium oxide ($Al_2O_3$; 5.59–16.2 wt%), FeO (2.79–24.4 wt%) and calcium oxide (CaO; 0.72–15.6 wt%) abundances (Supplementary Table 2). In the diagram of MgO versus $SiO_2$, $Al_2O_3$, FeO, CaO and sodium oxide ($Na_2O$; Supplementary Fig. 5), the melt inclusions plot close to the low-MgO endmember, consistent with the datasets of melt inclusions in olivine and pyroxene from Apollo basalts reported by ref. [11]. The apatite grains contain 2.35–3.28 wt% F and 0.11–0.87 wt% Cl (Supplementary Table 2), plotting close to the fluorapatite end-member in the F–Cl–OH ternary diagram (Supplementary Fig. 6). The apatite OH contents calculated by difference, assuming that the volatile site only contains F, Cl and OH, range from about 0 to 0.24 wt%.

**Estimate of water abundance for the parent magma of CE5 basalts**
**The melt inclusions in ilmenite.** The water abundances and δD values of melt inclusions were first corrected for the cosmic-ray spallation effects (Extended Data Table 3). After the spallation correction, the measured water abundances of the melt inclusions were corrected for post-entrapment crystallisation (PEC) based on the percentage of crystallized periphery of the melt inclusions (Extended Data Table 3). The water abundances of the melt inclusions were reduced by 0–52% by PEC correction. The water abundances and δD values of melt inclusions corrected for both spallation and PEC effects are summarized in Extended Data Table 3. Three melt inclusions were analysed twice, with two of them (clast numbers 103-020,013 and 103-020,018) showing larger variation in water content between repeated measurements. This observation could be attributed to partial covering on the re-crystallized materials in the melt inclusions.

The melt inclusions analysed in this work show two distinct populations in terms of water abundance and δD values (Fig. 2). Eight analyses on five individual basalt clasts define a negative correlation (main trend) between the water abundances (6 ± 2 µg g$^{-1}$ to 283 ± 22 µg g$^{-1}$) and δD values (−330 ± 190‰ to 200 ± 390‰) (Extended Data Table 3), obviously distinct from the analyses of the apatite grains in CE5 basalts (Fig. 2) and Apollo mare basalts[15,17,29,34,35,37]. The other three analyses of the ilmenite-hosted melt inclusions located in three other basalt clasts are more deuterium enriched (271 ± 124‰ to 869 ± 230‰) and contain 93 ± 15 µg g$^{-1}$ to 370 ± 21 µg g$^{-1}$ water, plotting in the region between the analyses of apatite and the negative trend of the melt inclusions (Fig. 2). The negative correlation between water abundances and δD values of the melt inclusions (δD < 200‰) can be explained by degassing of $H_2$ in the basaltic magma and deuterium enrichment during this process[11,15]. The water-rich and deuterium-poor end-member of these melt inclusions have δD values of −330 ± 190‰, within the range of the hydrogen isotopic compositions of the lunar mantle (about 0 ± 200‰) constrained by numerous measurements of Apollo melt inclusions[11], anorthosite[24], and apatite from KREEP basalts[16,61], high-Al basalts[29,36] and highlands samples[19,62,63]. The most deuterium-depleted melt inclusions probably captured the parent magma of CE5 basalts before significant degassing of $H_2$. Therefore, the water abundance of 283 ± 22 µg g$^{-1}$ of the deuterium-depleted melt inclusions can be regarded as the maximum water abundance estimate for the parent magma of CE5 basalts (Extended Data Table 3).

**Apatite.** Apatite is the major OH-, F- and Cl-bearing phase in lunar and other extraterrestrial samples. It was once widely used to estimate the water contents of the mantle reservoirs of Mars[64] and the Moon[13,14]. Recent numerical models have revealed that it is possible to crystallize

water-rich apatite from a highly water-depleted magma, because of the fractional crystallization and exchange behaviour of OH, F and Cl in apatite[65]. To calculate the water content of a silicate melt using the composition of coexisting apatite, many parameters are required, including the apatite-based melt hygrometry, the water content of the apatite, the apatite-melt exchange coefficient, the abundance of F or Cl in the apatite, and that of F or Cl in the melt[66]. However, it is difficult to determine all of these parameters precisely in the case of CE5 apatite.

Instead, we estimated the water abundance of the bulk CE5 basalts from the modal abundance of apatite and its average water content, because apatite is the dominant water-bearing phase in mare basalts. The modal abundance of apatite in the CE5 basalts is determined to be approximately 0.4 vol%, using the surface areas of apatite in all basaltic clasts analysed (Supplementary Table 1). The average water content and δD value of the CE5 apatite measured by NanoSIMS 50L are $1,921 \pm 910\ \mu g\ g^{-1}$ and $578 \pm 208‰$ ($N = 40$), respectively, except for the three low water abundance analyses (Extended Data Table 2). Hence, the water abundance of the bulk CE5 basalt is $7 \pm 3\ \mu g\ g^{-1}$. As discussed above, the parent magma of CE5 basalts has the original δD value of approximately −330‰ indicated by the most deuterium-depleted melt inclusions, whereas the deuterium enrichment of apatite was probably attributed to degassing of water in the form of $H_2$. Hence, the water abundance of the parent magma was calibrated to be $600 \pm 400\ \mu g\ g^{-1}$, with 98–99% water degassing loss required to enhance δD values from about −330‰ to about 600‰ based on the $H_2$ degassing modelling[41] (Extended Data Table 3). This estimate is consistent with the water abundance of the most deuterium-depleted melt inclusions in ilmenite.

**Clinopyroxene.** A higher $Cs^+$ primary beam current (7 nA) was used to analyse nominally anhydrous clinopyroxene, and the $H_2O$ background was reduced to $5 \pm 2\ \mu g\ g^{-1}$. Eighteen analyses on clinopyroxene from 13 individual CE5 basalt clasts yielded an average $^1H/^{16}O$ ratio of $8.85 \times 10^{-7}$, significantly lower than that of San Carlos olivine ($2.30 \times 10^{-6}$) measured under the same conditions (Extended Data Table 4). Therefore, all hydrogen emitted during CE5 clinopyroxene analyses can be ascribed to background hydrogen, indicating that the CE5 clinopyroxene contains less than $5\ \mu g\ g^{-1}\ H_2O$. The water abundance of the parent magma equilibrated with the CE5 clinopyroxene could be less than about $170\ \mu g\ g^{-1}$ using the water partition coefficient of 0.03 that was determined experimentally under reduced lunar conditions[33].

### Estimate of water abundance for the mantle source of CE5 basalts

The ilmenite-hosted melt inclusions were used to estimate the water abundance for the mantle source of CE5 basalts because they are the quenched parent melt and have preserved the original δD value of the mantle source. On the basis of the petrogenesis and geochemistry of CE5 basalts, the parent magma was derived from a depleted mantle source[4]. Furthermore, the REE patterns of the bulk CE5 basalts indicate that these basalts formed through low-degree (2–3%) partial melting followed by moderate-to-high-degree (43–88%) fractional crystallization[4]. Accordingly, a maximum water abundance of $1–5\ \mu g\ g^{-1}$ can be estimated for the depleted mantle source, which produced $283 \pm 22\ \mu g\ g^{-1}$ water in the derived parent magma of the CE5 basalts, assuming that all water partitioned into melt during partial melting of lunar mantle[33].

### Data availability

All geochemical data generated in this study are included in Extended Data Tables 1–5 and in Supplementary Tables 1–5, and are available on Zenodo at https://doi.org/10.5281/zenodo.5341793.

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

**Acknowledgements** We thank D. Chew for providing Durango and Kovdor apatite; E. Hauri for providing basaltic glass 519-4-1 and 1833-11; R. Francis for providing SWIFT MORB glass; R. Mitchell, X.-H. Li and F.-Y. Wu for constructive comments; Y. Chen and L. Jia for the assistance on EPMA measurement; W. Yang, H. Tian, H. Ma and D. Zhang for hand picking the CE5 soil fragments; and J. Yuan and X. Tang for the assistance with the SEM observation. This study was funded by the Strategic Priority Research Program of Chinese Academy of Sciences

(XDB 41000000), the key research programme of the Chinese Academy of Sciences (ZDBS-SSW-JSC007-15) and the key research programme of the Institute of Geology and Geophysics, Chinese Academy of Sciences (IGGCAS-202101 and 201904), the National Natural Science Foundation of China (41973062) and Pre-research project on Civil Aerospace Technologies by CNSA (D020201, D020203, and D020205). M.A. and R.T. acknowledge funding from the UK Science and Technology Facilities Council (grant numbers ST/P000657/1 and ST/P005225/1, respectively). The CE5 samples were allocated by the China National Space Administration.

**Author contributions** S.H., Y.L. and H. Hui designed this research. J.J., Huicun He, Y.Y., L.G., Q.G. and S.H. prepared the sample and characterized the petrography and mineral chemistry of CE5 basalts. Huicun He, J.H., R.L., J.J. and S.H. conducted the NanoSIMS measurements. S.H., Y.L., H. Hui, J.J., Huicun He, M.A. and R.T. wrote the manuscript. All authors contributed to the preparation of the manuscript.

**Competing interests** The authors declare no competing interests.

**Additional information**
**Correspondence and requests for materials** should be addressed to Sen Hu or Yangting Lin.

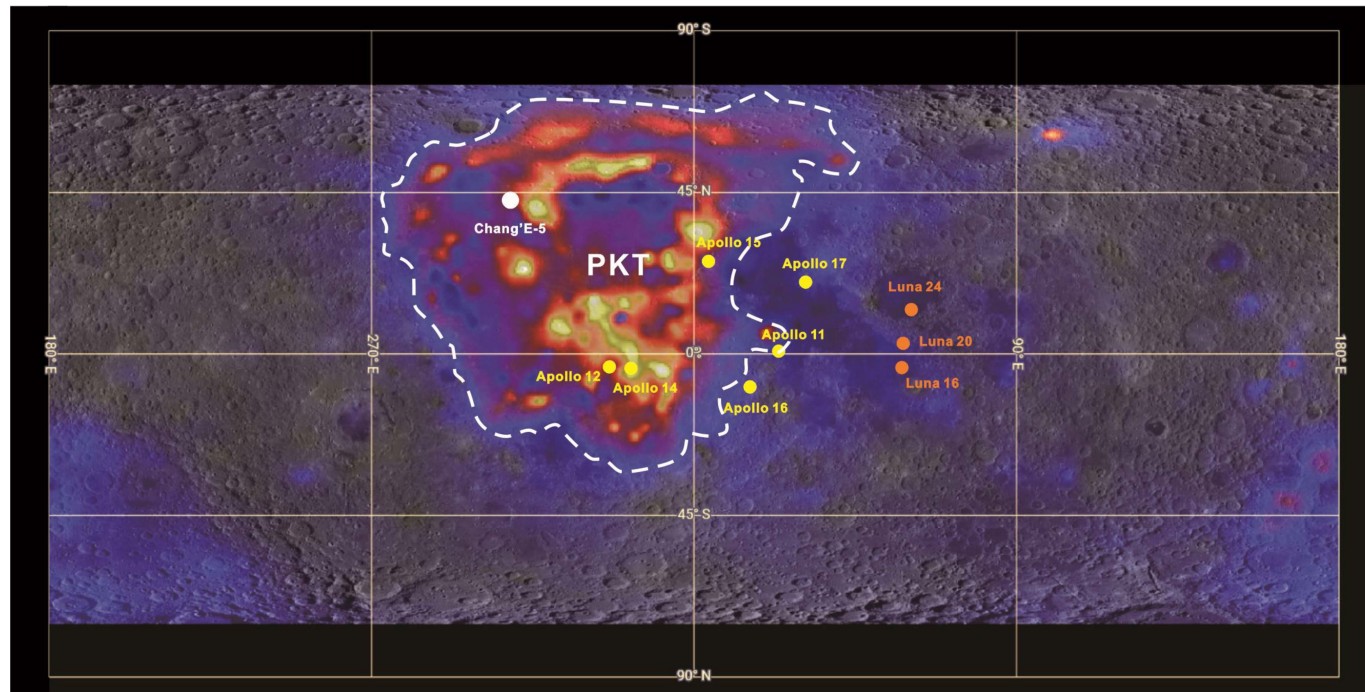

**Extended Data Fig. 1 | Location of CE5 samples on the Moon.** CE5 landed in a region with a high Th abundance within the northwestern Oceanus PKT (outlined) on 17 December 2020 (refs. [31,67].), far from the landing sites of Apollo and Luna missions. The map (LROC WAC Basemap) and the overlapping Th distribution (Lunar Prospector) are adapted from ref. [68].

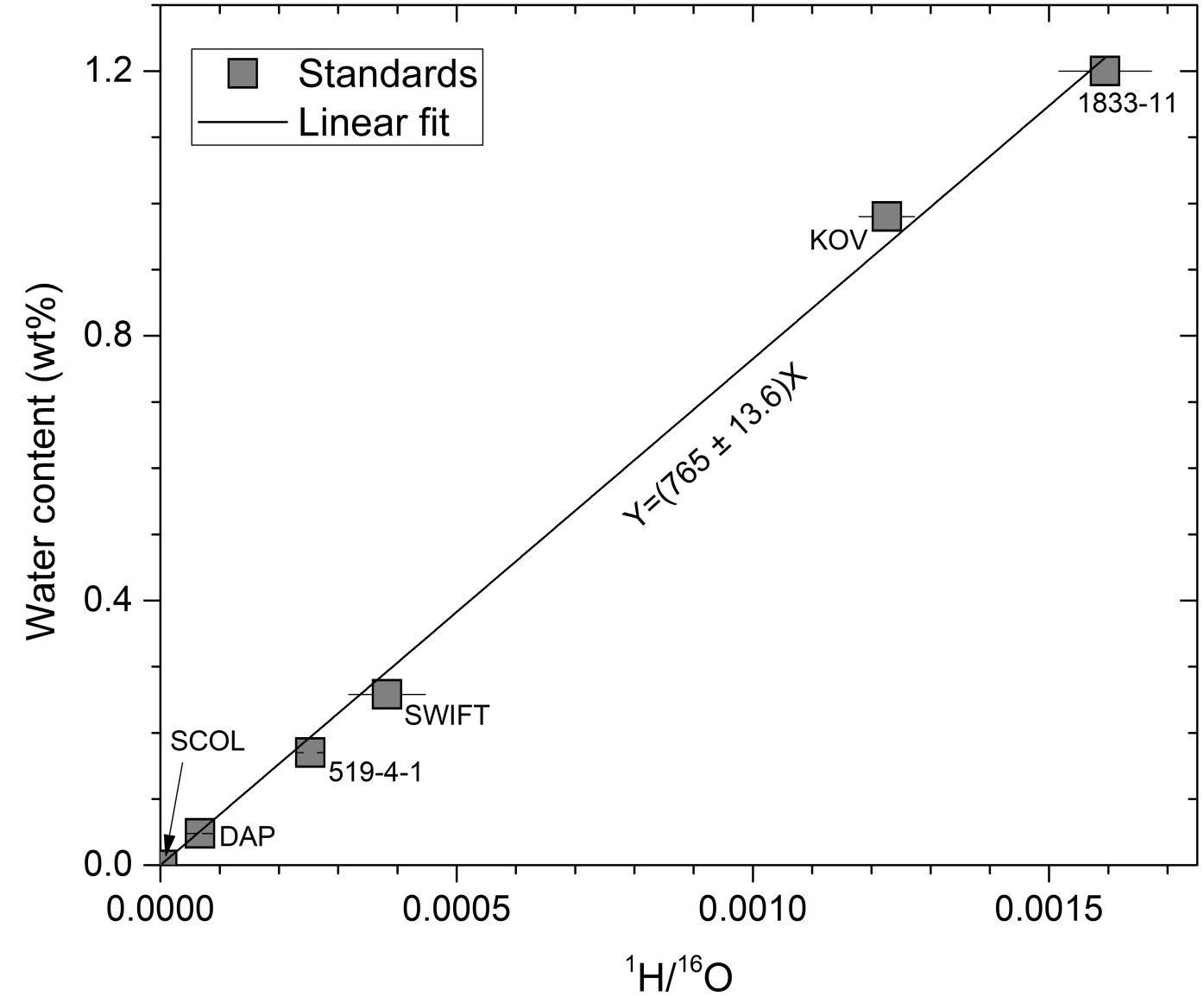

**Extended Data Fig. 2 | The water content calibration line established on apatite and silicate glass standards.** KOV, Kovdor apatite; DAP, Durango apatite; SWIFT, SWIFT MORB glass; 519-4-1, basaltic glass; 1833-11, basaltic glass; SCOL, San Carlos olivine. The datasets are listed in Supplementary Table 3. The analytical uncertainty is 0.6% ($2\sigma$).

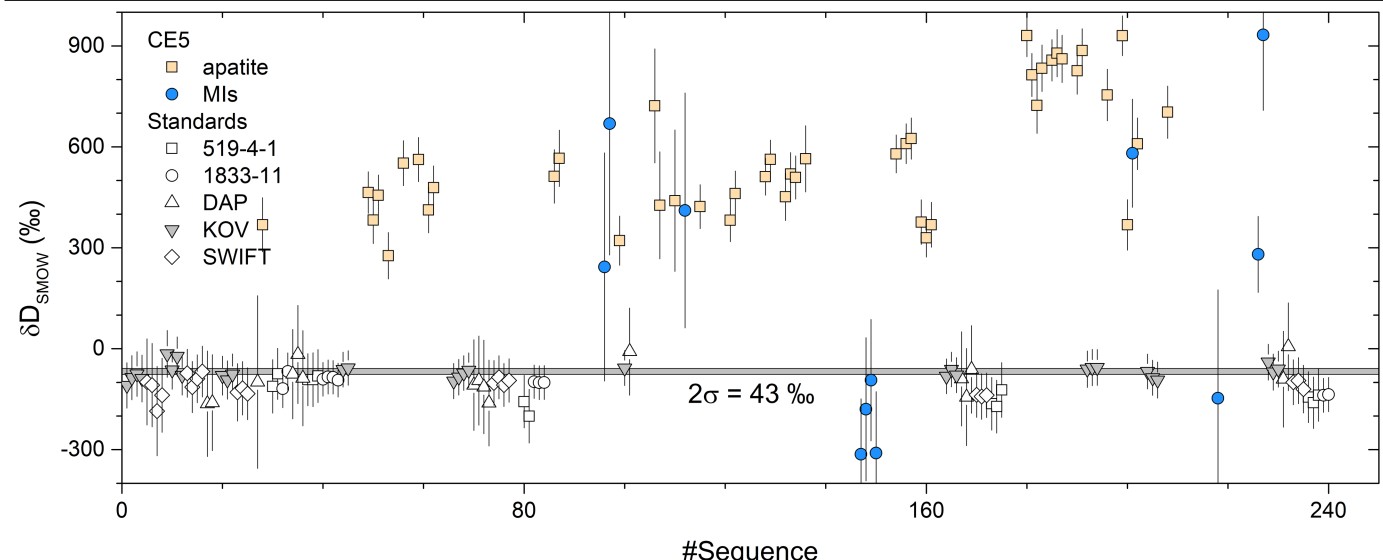

**Extended Data Fig. 3 | Reproducibility of hydrogen isotope analysis of the standards and the CE5 apatite and ilmenite-hosted melt inclusions over the analytical session.** The reproducibility of δD analysis throughout the whole analytical sessions was ±43‰ (2σ), estimated on the Kovdor apatite standard. The average δD values measured for the SWIFT MORB glass and Durango apatite are −113 ± 109‰ and −92 ± 210‰, respectively, consistent with their recommended values[29,57] within analytical errors. The average δD values measured for the basaltic glass standards 1833-11 and 519-4-1 are −101 ± 86‰ and −132 ± 158‰, respectively. All analytical data are listed in Extended Data Tables 3, 4, Supplementary Table 3.

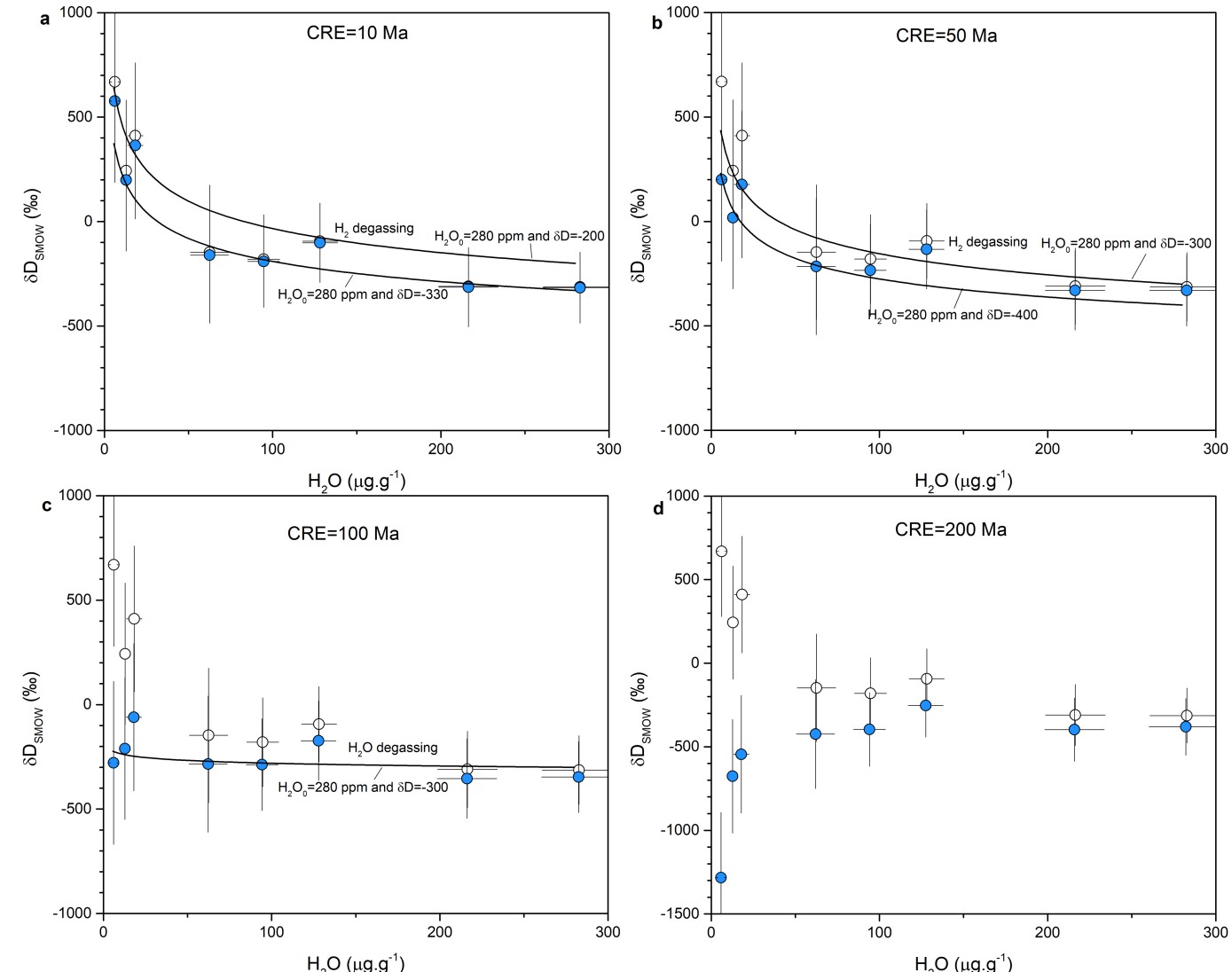

**Extended Data Fig. 4 | Spallation correction modelling for ilmenite-hosted melt inclusions after PEC correction.** A cosmogenic deuterium production rate of $2.17 \times 10^{-12}$ mol D $g^{-1}$ $Ma^{-1}$ (ref. [59]) was used for melt inclusion glasses. (a) CRE age = 10 Ma. (b) CRE age = 50 Ma. (c) CRE age =100 Ma. (d) CRE age = 200 Ma. Open circles correspond to measured data and blue filled circles correspond to CRE-corrected data (Extended Data Table 4, Supplementary Table 4). Modelled degassing curves have been calculated assuming Rayleigh fractionation into a vacuum, using masses of 2.016 and 3.022, and a fractionation factor of $(2.016/3.022)^{1/2}$ for $H_2$, and masses of 18.015 and 19.021, corresponding to a fractionation factor of $(18.015/19.021)^{1/2}$ for $H_2O$, following the procedure of ref. [41] (Methods).

## Extended Data Table 1 | The CE5 basalt clasts

| Clast No. | Mount form* | Dimension (mm) | Texture | Apatite (μm) | Melt inclusion (μm) |
|---|---|---|---|---|---|
| CE5C0100YJFM00406 | | | | | |
| 406-010,001 | Sn-Bi alloy | 0.88×0.39 | Poikilitic | ~3 | |
| 406-010,019 | Sn-Bi alloy | 0.56×0.36 | Poikilitic | ~3 | |
| 406-010,023 | Sn-Bi alloy | 0.78×0.58 | Poikilitic | 10 | 17×11 |
| 406-011,003 | Sn-Bi alloy | 0.88×0.61 | Equigranular | 5 to 10 | |
| 406-011,007 | Sn-Bi alloy | 0.55×0.32 | Poikilitic | 3 to 5 | |
| 406-012,004 | Sn-Bi alloy | 0.86×0.64 | Subophitic | 3 to 5 | |
| 406-012,009 | Sn-Bi alloy | 0.70×0.53 | Fragment | ~3 | <50×17 |
| 406-014,001 | PTS | 0.40×0.19 | Fragment | <10 | |
| 406-014,003 | PTS | 0.73×0.66 | Subophitic | ~5 | |
| 406-015,001 | PTS | 0.26×0.19 | Fragment | 10 | |
| 406-015,014 | PTS | 0.51×0.41 | Equigranular | 10 | |
| 406-015,045 | PTS | 0.72×0.30 | Subophitic | 3 to 5 | 19×7 |
| 406-015,046 | PTS | 0.60×0.33 | Subophitic | 3 to 5 | 6×4 |
| 406-015,048 | PTS | 0.65×0.30 | Subophitic | 5 to 10 | |
| 406-015,059 | PTS | 0.41×0.29 | Poikilitic | 3 to 5 | |
| CE5C0100YJFM00103 | | | | | |
| 103-017,001 | Sn-Bi alloy | 1.03×0.41 | Poikilitic | ~3 | 55×40 |
| 103-017,010 | Sn-Bi alloy | 0.44×0.32 | Equigranular | ~5 | |
| 103-017,011 | Sn-Bi alloy | 0.63×0.45 | Equigranular | ~5 | |
| 103-017,013 | Sn-Bi alloy | 0.54×0.44 | Poikilitic | ~5 | 10×7 |
| 103-020,004 | PTS | 0.28×0.26 | Fragment | 10 | |
| 103-020,013 | PTS | 0.37×0.26 | Fragment | | 25×13 |
| 103-020,018 | PTS | 0.42×0.30 | Fragment | <3 | 17×4 |
| 103-020,021 | PTS | 0.39×0.23 | Poikilitic | 5 to 10 | |

*Sn–Bi alloy mount; PTS, polished thin section.

# Extended Data Table 2 | Water abundance and hydrogen isotopes of CE5 apatite

| File Name | Clast No. | CRE age = 0 Ma | | | | CRE age = 50 Ma[*] | | | |
|---|---|---|---|---|---|---|---|---|---|
| | | $H_2O$ µg.g$^{-1}$ | 2σ µg.g$^{-1}$ | δD ‰ | 2σ ‰ | $H_2O$ µg.g$^{-1}$ | 2σ µg.g$^{-1\#}$ | δD ‰ | 2σ ‰[†] |
| 010-001Ap-1 | 406-010,001 | 2205 | 91 | 382 | 64 | 2205 | 91 | 381 | 81 |
| 010-001Ap-2 | 406-010,001 | 1850 | 79 | 461 | 67 | 1850 | 79 | 460 | 84 |
| 010-019Ap-1 | 406-010,019 | 2418 | 109 | 422 | 65 | 2418 | 109 | 421 | 82 |
| 010-019Ap-2 | 406-010,019 | 2153 | 96 | 363 | 64 | 2153 | 96 | 362 | 81 |
| 010-023Ap-1 | 406-010,023 | 176 | 17 | 722 | 169 | 176 | 17 | 708 | 176 |
| 010-023Ap-2 | 406-010,023 | 236 | 19 | 426 | 159 | 235 | 19 | 416 | 167 |
| 010-023Ap-3 | 406-010,023 | 110 | 13 | 440 | 210 | 110 | 13 | 418 | 216 |
| 011-003Ap-1 | 406-011,003 | 2936 | 119 | 464 | 62 | 2936 | 119 | 463 | 80 |
| 011-007Ap-1 | 406-011,007 | 1626 | 73 | 276 | 69 | 1626 | 73 | 275 | 85 |
| 012-004Ap-1 | 406-012,004 | 2595 | 108 | 382 | 70 | 2595 | 108 | 381 | 86 |
| 012-004Ap-2 | 406-012,004 | 2755 | 110 | 456 | 60 | 2755 | 110 | 455 | 78 |
| 012-009Ap-1 | 406-012,009 | 1936 | 87 | 551 | 67 | 1936 | 87 | 550 | 84 |
| 014-001Ap-1 | 406-014,001 | 4363 | 187 | 329 | 57 | 4363 | 187 | 329 | 76 |
| 014-001Ap-2 | 406-014,001 | 1841 | 82 | 368 | 67 | 1841 | 82 | 367 | 84 |
| 014-003Ap-1 | 406-014,003 | 4856 | 217 | 376 | 66 | 4856 | 217 | 376 | 83 |
| 015-001Ap-1 | 406-015,001 | 1137 | 53 | 368 | 80 | 1137 | 53 | 366 | 94 |
| 015-001Ap-2 | 406-015,001 | 1167 | 56 | 512 | 80 | 1166 | 56 | 510 | 94 |
| 015-001Ap-3 | 406-015,001 | 939 | 46 | 566 | 84 | 939 | 46 | 563 | 98 |
| 015-014Ap-1 | 406-015,014 | 1717 | 74 | 931 | 63 | 1717 | 74 | 929 | 80 |
| 015-014Ap-2 | 406-015,014 | 1652 | 71 | 814 | 64 | 1651 | 71 | 812 | 81 |
| 015-014Ap-3 | 406-015,014 | 791 | 40 | 723 | 83 | 791 | 40 | 720 | 97 |
| 015-014Ap-4 | 406-015,014 | 1461 | 64 | 834 | 69 | 1461 | 64 | 832 | 85 |
| 015-014Ap-5 | 406-015,014 | 1983 | 88 | 857 | 61 | 1983 | 88 | 856 | 79 |
| 015-014Ap-6 | 406-015,014 | 1214 | 54 | 878 | 70 | 1213 | 54 | 877 | 86 |
| 015-014Ap-7 | 406-015,014 | 1139 | 52 | 861 | 70 | 1138 | 52 | 859 | 86 |
| 015-014Ap-8 | 406-015,014 | 1312 | 59 | 1024 | 71 | 1312 | 59 | 1022 | 87 |
| 015-014Ap-9 | 406-015,014 | 1217 | 54 | 826 | 70 | 1217 | 54 | 824 | 86 |
| 015-014Ap-10 | 406-015,014 | 1516 | 67 | 886 | 65 | 1516 | 67 | 884 | 82 |
| 015-014Ap-11 | 406-015,014 | 2361 | 98 | 930 | 59 | 2361 | 98 | 929 | 77 |
| 015-014Ap-12 | 406-015,014 | 952 | 45 | 703 | 78 | 952 | 45 | 700 | 93 |
| 015-046Ap-1 | 406-015,046 | 1574 | 69 | 368 | 75 | 1574 | 69 | 367 | 90 |
| 015-048Ap-1 | 406-015,048 | 1127 | 65 | 609 | 77 | 1127 | 65 | 607 | 92 |
| 015-059Ap-1 | 406-015,059 | 1166 | 57 | 754 | 77 | 1166 | 57 | 752 | 92 |
| 017-010Ap-1 | 103-017,010 | 1606 | 75 | 321 | 73 | 1606 | 75 | 320 | 88 |
| 017-011Ap-1 | 103-017,011 | 1714 | 80 | 412 | 68 | 1714 | 80 | 411 | 84 |
| 017-011Ap-2 | 103-017,011 | 2322 | 96 | 479 | 65 | 2322 | 96 | 478 | 82 |
| 017-013Ap-1 | 103-017,013 | 1920 | 87 | 562 | 66 | 1920 | 87 | 561 | 83 |
| 020-004Ap-1 | 103-020,004 | 1807 | 82 | 451 | 70 | 1807 | 82 | 450 | 86 |
| 020-004Ap-2 | 103-020,004 | 2164 | 91 | 519 | 64 | 2164 | 91 | 518 | 81 |
| 020-004Ap-3 | 103-020,004 | 2049 | 93 | 509 | 64 | 2049 | 93 | 508 | 81 |
| 020-018Ap-1 | 103-020,018 | 555 | 31 | 564 | 98 | 555 | 31 | 560 | 110 |
| 020-021Ap-1 | 103-020,021 | 3681 | 169 | 511 | 55 | 3681 | 169 | 511 | 74 |
| 020-021Ap-2 | 103-020,021 | 3077 | 134 | 563 | 58 | 3077 | 134 | 562 | 77 |

[*]Deuterium production rate of $9.20 \times 10^{-13}$ mol D g$^{-1}$ Ma$^{-1}$ (ref. [60]) was used for correction of cosmogenic spallation effects. CRE age, cosmic ray exposure age.

[#]Spallation correction of water content is 0.18 ppm, the increased uncertainty on water abundance is ignored.

[†]Uncertainties include another 50‰ caused by spallation correction[21].

**Extended Data Table 3 | Water abundance and hydrogen isotopes of CE5 ilmenite-hosted melt inclusions**

| File Name | Clast No. | CRE age = 0 Ma | | | | CRE age = 50 Ma[*] | | | | PEC corrected | | |
|---|---|---|---|---|---|---|---|---|---|---|---|---|
| | | $H_2O$ μg.g$^{-1}$ | 2σ μg.g$^{-1}$ | δD ‰ | 2σ ‰ | $H_2O$ μg.g$^{-1}$ | 2σ μg.g$^{-1#}$ | δD ‰ | 2σ ‰$^{†}$ | $H_2O$ μg.g$^{-1}$ | 2σ μg.g$^{-1}$ | PEC %$^{‡}$ |
| 010-023MI-1 | 406-010,023 | 26 | 6 | 411 | 349 | 26 | 6 | 177 | 352 | 18 | 5 | 29 |
| 012-009MI-1 | 406-012,009 | 89 | 16 | -147 | 322 | 89 | 16 | -216 | 326 | 62 | 11 | 30 |
| 015-045MI-1 | 406-015,045 | 661 | 37 | 281 | 113 | 661 | 37 | 271 | 124 | 370 | 21 | 44 |
| 015-046MI-1 | 406-015,046 | 174 | 18 | 581 | 161 | 174 | 18 | 547 | 170 | 125 | 13 | 28 |
| 017-013MI-1 | 406-017,013 | 93 | 15 | 933 | 224 | 93 | 15 | 869 | 230 | 93 | 15 | 0 |
| 017-001MI-1 | 406-017,001 | 27 | 6 | 243 | 339 | 27 | 6 | 18 | 340 | 13 | 3 | 52 |
| 017-001MI-2 | 406-017,001 | 13 | 4 | 669 | 390 | 13 | 4 | 200 | 390 | 6 | 2 | 52 |
| 020-013MI-1 | 103-020,013 | 367 | 29 | -313 | 164 | 367 | 29 | -330 | 170 | 283 | 22 | 23 |
| 020-013MI-2 | 103-020,013 | 281 | 23 | -310 | 182 | 281 | 23 | -330 | 190 | 216 | 18 | 23 |
| 020-018MI-1 | 103-020,018 | 114 | 11 | -180 | 213 | 114 | 11 | -234 | 220 | 94 | 9 | 17 |
| 020-018MI-2 | 103-020,018 | 154 | 13 | -94 | 180 | 154 | 13 | -133 | 190 | 128 | 10 | 17 |

[*]Deuterium production rate of $2.17 \times 10^{-12}$ mol D g$^{-1}$ Ma$^{-1}$ (ref. [59]) was used for correction of cosmogenic spallation effects. CRE age, cosmic ray exposure age.

[#]Spallation correction of water content is 0.18 ppm, the increased uncertainty on water abundance is ignored.

[†]Uncertainties includes another 50‰ caused by spallation correction[21].

[‡]Post-entrapment crystallization (PEC) is calculated from the areas of the BSE images.

**Extended Data Table 4 | H/O ratios of CE5 clinopyroxene and reference San Carlos olivine measured by NanoSIMS 50L**

| File Name | Clast No. | $^1H_{Counts}$ | $^{16}O_{Counts}$ | $^1H/^{16}O$ | Err Mean % | Poisson % |
|---|---|---|---|---|---|---|
| SanCarlosOl-1 | Standard | 1.39E+05 | 5.40E+10 | 2.58E-06 | 8.50E-01 | 2.68E-01 |
| SanCarlosOl-2 | Standard | 1.19E+05 | 4.65E+10 | 2.56E-06 | 1.15E+00 | 2.90E-01 |
| SanCarlosOl-3 | Standard | 1.12E+05 | 4.13E+10 | 2.71E-06 | 8.36E-01 | 2.99E-01 |
| SanCarlosOl-4 | Standard | 6.21E+04 | 4.62E+10 | 1.34E-06 | 7.49E-01 | 4.01E-01 |
| *Avg.* | | | | *2.30E-06* | | |
| 10-019CPX-1 | 406-010,019 | 4.60E+04 | 7.55E+10 | 6.09E-07 | 6.74E-01 | 4.66E-01 |
| 10-019CPX-2 | 406-010,019 | 3.72E+04 | 6.61E+10 | 5.62E-07 | 6.84E-01 | 5.19E-01 |
| 10-019CPX-3 | 406-010,019 | 3.72E+04 | 6.91E+10 | 5.39E-07 | 7.20E-01 | 5.18E-01 |
| 10-023CPX-1 | 406-010,023 | 4.61E+04 | 6.49E+10 | 7.11E-07 | 7.98E-01 | 4.66E-01 |
| 10-023CPX-2 | 406-010,023 | 4.01E+04 | 6.87E+10 | 5.84E-07 | 6.32E-01 | 4.99E-01 |
| 12-009CPX-1 | 406-012,009 | 3.69E+04 | 6.13E+10 | 6.02E-07 | 7.83E-01 | 5.20E-01 |
| 16-005CPX-1 | 406-016,005 | 4.79E+04 | 7.39E+10 | 6.49E-07 | 5.15E-01 | 4.57E-01 |
| 16-011CPX-1 | 406-016,011 | 7.20E+04 | 5.63E+10 | 1.28E-06 | 5.99E-01 | 3.73E-01 |
| 16-013CPX-1 | 406-016,013 | 5.85E+04 | 5.46E+10 | 1.07E-06 | 8.24E-01 | 4.13E-01 |
| 16-017CPX-1 | 406-016,017 | 5.11E+04 | 4.95E+10 | 1.03E-06 | 8.82E-01 | 4.42E-01 |
| 16-017CPX-2 | 406-016,017 | 5.78E+04 | 5.46E+10 | 1.06E-06 | 8.26E-01 | 4.16E-01 |
| 17-010CPX-1 | 103-017,010 | 5.07E+04 | 5.30E+10 | 9.56E-07 | 1.17E+00 | 4.44E-01 |
| 18-001CPX-1 | 103-018,001 | 4.44E+04 | 5.92E+10 | 7.50E-07 | 6.45E-01 | 4.75E-01 |
| 18-005CPX-1 | 103-018,005 | 4.74E+04 | 6.11E+10 | 7.76E-07 | 5.61E-01 | 4.60E-01 |
| 20-001CPX-1 | 103-020,001 | 7.99E+04 | 6.41E+10 | 1.25E-06 | 4.66E-01 | 3.54E-01 |
| 20-013CPX-1 | 103-020,013 | 6.37E+04 | 5.12E+10 | 1.24E-06 | 7.20E-01 | 3.96E-01 |
| 20-018CPX-1 | 103-020,018 | 6.81E+04 | 6.03E+10 | 1.13E-06 | 5.15E-01 | 3.83E-01 |
| 20-018CPX-2 | 103-020,018 | 6.58E+04 | 5.82E+10 | 1.13E-06 | 8.08E-01 | 3.90E-01 |
| *Avg.* | | | | *8.85E-07* | | |

**Extended Data Table 5 | Summary of the water abundances estimated for the lunar mantle source regions of basaltic products formed between around 4 Ga and 2 Ga**

| Sample name | Age (Ga)[*] | References | Phase[#] | $H_2O_{min}$ ($\mu g.g^{-1}$) | $H_2O_{max}$ ($\mu g.g^{-1}$) | References |
|---|---|---|---|---|---|---|
| 10020 | 3.7 | [69] | MI | 3 | 60 | [11] |
| 10044 | 3.71 | [70] | Ap | 2.12 | 3.12 | [29, 35] |
| 10058 | 3.63 | [70] | Ap | 9 | 28 | [15] |
| 10058 | 3.63 | [70] | MI | 3 | 60 | [11] |
| 12002 | 3.36 | [70] | MI | 3 | 3 | [11] |
| 12004 | 3.36 | [70] | MI | 3 | 3 | [11] |
| 12008 | 3.36 | [70] | MI | 3 | 3 | [11] |
| 12018 | ND | | MI | 25 | 160 | [11, 37] |
| 12020 | 3.36 | [70] | MI | 3 | 3 | [11] |
| 12039 | 3.19 | [71] | Ap | 2.4 | 15.12 | [15, 28, 29, 34] |
| 12040 | 3.36 | [70] | MI | 3 | 3 | [11] |
| 12064 | 3.18 | [72] | Ap | 2 | 7 | [25, 35] |
| 14053 | 3.96 | [72] | Ap | 1.42 | 26 | [29, 36, 73] |
| 14072 | 3.99 | [6] | MI | 3 | 60 | [11] |
| 15016 | 3.34 | [74] | MI | 3 | 60 | [11] |
| 15058 | 3.36 | [75] | Ap | 1 | 4 | [14, 15] |
| 15427 | 3.41 | [76] | GB | 5.2 | 14.9 | [73] |
| 15555 | 3.31 | [77] | Ap | 10 | 30 | [15] |
| 74002 | 3.66 | [78] | GB | 4 | 92 | [79] |
| 74220 | 3.48 | [80] | MI | 9 | 130 | [8, 25] |
| 74220 | 3.48 | [80] | MI | 133 | 292 | [22] |
| 74220 | 3.48 | [80] | MI | 88 | 124 | [9] |
| 74235 | 3.48 | [80] | MI | 84 | 84 | [10] |
| 75055 | 3.77 | [81] | Ap | 4 | 13 | [28, 29, 34] |
| NWA 2977 | 2.86 | [5] | Ap | 7.2 | 17 | [14] |
| MIL 05035 | 3.85 | [82] | Ap | 8 | 26 | [15] |
| LAP 04841 | 3.0 | [83] | Ap | 53 | 166 | [15] |
| CE5 | 2.03 | [3] | MI | 1 | 5 | This study |

Data are from refs. [3,5,6,8–11,14,15,22,25,28,29,34–37,69–83].

*ND, no data.

#Phase used for estimating water abundance in the lunar mantle: MI, melt inclusions; Ap, apatite; GB, glass bead.