## [Peer Review File · Nature]

Manuscript Title: A dry lunar mantle reservoir for young mare basalts of Chang'E-5

Editorial Note

Redactions-- unpublished data

Reviewer Comments & Author Rebuttals

Reviewer Reports on the Initial Version:

Referee #1:

The distribution of water in the lunar interior and its role in mare magmatism has significant implications for our understanding of the Moon and its evolution. This manuscript is important in that it examines the volatile content of the youngest mare basalts thus far returned from the Moon and the volatile content of highly depleted mare basalt sources. This manuscript is highly appropriate for Nature.

Other general comments:

The melt inclusions and apatite are placed within a petrologic context that makes the interpretation of the H data easier.

The manuscript provides a reasonable approach for the presentation of the OH in apatite and correction of D/H for due to surface interactions with cosmic ray spallation. Although CRE ages have yet to be determined for these basalts, the rationale behind the spallation correction is sound.

Line 37 "Dried up" seems a bit awkward for this manuscript. Dehydrated could be a better term to use. Prior melt extraction is one manner to cause dehydration of the source. This potentially fits the depleted nature of the source. Alternatively, the source (LMO cumulates) may have originally been low in the primordial H component.

Lines 49-60 In addition to all of these factors, there is probably an issue based on the assumptions of the calculations of the mantle source. For example, how much degassing occurred prior to the crystallization of apatite or during fire fountaining, how much crystallization occurred following extraction of the basalt from its source, how much melting occurred. Please don't include all of these possibilities, however it is important to simply insert errors based on calculation assumptions.

Lines 70-74. Please provide references for these statements. Further, there has been analyses of KREEPy magmatic lithologies for OH in apatite. Do these exhibit an abundant OH component?

Lines 152-154. This sentence does not adequately address the conclusion that little exchange of H between melt inclusions and ilmenite. Please expand this idea into an additional sentence.

Figure 2. The addition of Apollo samples on this diagram subtracts from the observations made on the CE5 samples. I suggest that only the CE 5 data and interpretation is kept in this diagram. A comparison to the Apollo sample data is presented in a separate diagram in an Appendix.

Lines 205-209. The model for producing an incompatible element enriched magma from a depleted source in the manuscript "Non-KREEP origin for Chang'E-5 basalt in the Procellarum KREEP Terrane" does not reproduce the REE pattern of the CE5 basalts. Therefore, the calculations here may have issues. I would suggest that the melting-crystallization model in Non-KREEP origin for Chang'E-5 basalt in the Procellarum KREEP Terrane is improved and then applied to this calculation. In this calculation, the liquid line of descent models by John Longhi could be used to confirm the modeled % crystallization after melt extraction.

Lines 231-233. This evolution of H content of the lunar mantle with time is not supported by the data in Figure 3. There appears to be numerous older basalts between 3.2 and 4.0 Ga that are

also "depleted" in the H component of their sources. This appears not to be time dependent but mantle source dependent (e.g., LMO cumulate horizon, or previous melting). Perhaps, the authors should investigate if the low H component in the older lunar basalts is partially tied to the depleted nature of the source.

Lines 241-248. It is an interesting premise that the H in the lunar mantle evolves with time. However, the data in Figure 3 does not support this model. The concept presented here is less exciting, but is more likely.

Lines 256-257. Perhaps this is not much of an enigma as implied here. This part of the Moon has high concentrations of heat producing elements. It is possible that the high heat-flow associated with this region of the Moon enabled the melting of enriched and depleted mantle sources.

Charles "Chip" Shearer

Referee #2:

Comments to Nature manuscript 2021-07-12319 "A dry lunar mantle reservoir for young mare basalts of Chang'E-5" by Hu et al.

Hu et al. report new 'water' contents and D/H ratios for apatite and ilmenite-hosted melt inclusions in Chang'E-5 basaltic clasts determined by SIMS. Since the CE5 mission recently sampled young mare units, far from the Apollo and Luna landing sites, these results may provide crucial new constraints on the spatial distribution and temporal evolution of the water abundance in the Moon's interior. Notably, the new results suggest that the water concentration in the mantle source sampled by C5 basalts is very low.

While I am excited by the topic and analytical approach of this study, I have critical concerns about the 'water' contents and spallation-corrected δD values reported for the (low-H₂O) melt inclusions, as detailed below. Consequently, I am not convinced that the 3-stage evolutionary path proposed for the melt (which is mainly based on MI data) is reliable. Due to the concerns I have about the measurements and data treatment, I am unable to support publication of the manuscript in its current form.

1) SIMS analyses:

1.1) 12 out of the 23 studied clasts were prepared as thin sections mounted in epoxy. This is not ideal for H analyses by SIMS. The authors should clarify if H backgrounds (i.e., H signals of San Carlos olivine) and H signals of pyroxene were comparable for samples mounted in epoxy or Sn-Bi. This is important for assessing the accuracy of low-H₂O analyses.

1.2) San Carlos olivine was used to determine the instrumental background of H.

The 'water' content of San Carlos is stated to be 1.4 ppm on Line 442. However, San Carlos olivine can contain 10's to 100's of ppm OH (see Kurosawa, M., Yurimoto, H. and Sueno, S. (1997), Phys Chem Min 24, 385), suggesting that it does not represent an appropriate dry standard for low-H₂O concentration measurements and/or instrumental background measurements. How can the authors be sure that their olivine grains contain only 1.4 ppm H₂O? This crucial issue must be addressed here.

The authors state that the H₂O background determined from SC analyses is 25 ± 8 ppm.

Nonetheless, it appears that the water content calibration line (Extended Data Fig. 2) was forced through zero. This suggests that the background correction is wrong. This would greatly affect the calculated water contents of low-H₂O samples (i.e., melt inclusions). Please clarify.

Also note that on Line 103 of the suppl. info, the reported instrument background of H is 5 ± 2 ppm. This is confusing.

1.3) Extended Data Fig. 2 suggests that one single water content calibration line was used that

includes both apatite and glass standards. This is unusual. Does this suggest that there is not matrix effect for glasses and apatite (and cpx)? Again, this may be important for low-H₂O analyses.

1.4) Extended Data Fig. 3 indicates that the measured δD values of the standard glass and apatite were "consistent with their recommended values". The lack of any instrumental mass fractionation is highly unusual. Does this imply that no correction of IMF was applied?

1.5) Fig. 1b and Fig. S1 indicate that many SIMS spots in melt inclusions included mineral phases (as opposed to a glass phase only). This is also confirmed by the EMPA results reported in Table S2. This implies that a correction needs to be applied to their H₂O content since the remaining melt will be enriched in H compared to the crystallized minerals. Notably, the inclusion that is considered to be the most primitive (103-020,013) appears to be entirely re-crystallized. This important issue must be addressed and discussed.

Could the high water content of MI 406-015,045 be explained by a contribution of 'water' in nearby fractures?

2) Correction of D/H ratios for spallation effects

The D production rate from Merlivat et al. (1976) was used to correct the D/H ratios of apatite. Why? This value was established for a basalt.

Hashizume et al. (EPSL, 2002) and Fűri et al. (GPL, 2018) showed that single grains within a regolith sample have highly variable exposure ages that can range up to 400 Ma (even up to >1Ga). Therefore, the use of a nominal CRE age of 50 Ma for all CE5 basalt clasts is unlikely to be correct. This suggests that the calculated D/H ratios of low-H₂O samples (i.e., melt inclusions) are wrong. Knowledge of the CRE age of each individual grain is necessary for an accurate spallation correction.

3) Melt inclusions in ilmenite - Water abundance of the CE5 mantle source

3.1) Three melt inclusions were analyzed twice (see Extended Data Table 3), and the measured water abundances and/or D/H ratios are highly variable. This observation should be discussed and explained.

3.2) "the water-richest melt inclusions have δD values of $\sim -300\text{‰}$ " (Supp. Info. Line 62) - this is not true! The water-richest MI has a δD value of $+271\text{‰}$. This misleading statement must be rephrased. (Please also rephrase the sentences on Lines 177 and 199-200 of the main text.)

3.3) Which water partition coefficient was used to derive the water content of the mantle source? Is this value well known for lunar mantle conditions? Please report these details in section "5.1." (should be 2.1?) of the suppl. info.

4) δD value of the CE5 mantle source

The parent magma (and mantle source) of CE5 basalts is estimated to have a δD value of $\sim -330\text{‰}$. Please briefly state what this value could indicate about the origin of water on the Moon. The CE5 mantle source is expected to have experienced multiple water-bearing melt extraction episodes (Lines 234-235). Are these melt extraction events expected to have modified the initial δD value of the mantle source?

Additional comments/questions:

Line 86: apatite is "F-rich and Cl-poor"?

Line 382: I presume the sections "were initially coated with carbon" for SEM analyses and then re-coated with Au. Please clarify.

Lines 412-413: It is not clear which ions were measured on EMs or a Faraday cup. Please rephrase.

Line 420: Please give the units (cps?) for Hbg.

Line 429: "more details can be found in Hu et al."

Extended Data Tables 2 and 3: The uncertainties of the δD values for CRE age = 50 Ma should be reported because these values are used in the discussion.

Extended Data Fig. 1, Line 580: "represented by a high Th abundance"? Please rephrase/clarify.

Extended Data Fig. 3: $\delta DSMOW$ - correct y-axis label

Supp. Info. Line 61: "supply with convincing evidence" - rephrase.

Supp. Info. Line 67: "pyroxene and olivine could have deposited" - rephrase.

Supp. Info. Line 78: "regardless large variation" - not clear, please rephrase.

Supp. Info. Line 100: "has be investigated" - replace by "has been found to be"

Table S1: Please explain what you mean by "size" and "total area".

(Maybe it would make more sense to report the areas in μm^2 .)

Table S2: The TiO_2 content of ilmenite is not correctly reported.

Table S3: Please check the format of the reported data (e.g. Err Mean of the 2H/1H ratio).

Which results are shown in the column labeled "Ratio"?

Author Rebuttals to Initial Comments:

Response to Referee #1

Referee #1:

The distribution of water in the lunar interior and its role in mare magmatism has significant implications for our understanding of the Moon and its evolution. This manuscript is important in that it examines the volatile content of the youngest mare basalts thus far returned from the Moon and the volatile content of highly depleted mare basalt sources. This manuscript is highly appropriate for Nature.

Other general comments:

The melt inclusions and apatite are placed within a petrologic context that makes the interpretation of the H data easier.

The manuscript provides a reasonable approach for the presentation of the OH in apatite and correction of D/H for due to surface interactions with cosmic ray spallation. Although CRE ages have yet to be determined for these basalts, the rationale behind the spallation correction is sound.

Line 37 “Dried up” seems a bit awkward for this manuscript. Dehydrated could be a better term to use. Prior melt extraction is one manner to cause dehydration of the source. This potentially fits the depleted nature of the source. Alternatively, the source (LMO cumulates) may have originally been low in the primordial H component.

Reply. As suggested, “Dried up” has been revised as “dehydrated”.

We agree that the lunar mantle sources (LMO cumulates as you mentioned) could originally be water-poor, and this scenario has been discussed as one possibility together with the dehydration model (Lines 208-213). However, we cannot further assess the original water abundance of the mantle source based on the youngest basalts returned by the CE5 basalts. Instead of young basalts, the original water abundances of the LMO cumulates can be estimated better using oldest basalts, because the water abundances of the mantle sources could have been modified by late partition melting. Hence, in the introductory paragraph, we propose that the low water abundance of the CE5 mantle source could be due to dehydration through previous melt extraction, which is consistent with the very young age of CE5 basalts.

Lines 49-60 In addition to all of these factors, there is probably an issue based on the assumptions of the calculations of the mantle source. For example, how much degassing occurred prior to the crystallization of apatite or during fire fountaining, how much crystallization occurred following extraction of the basalt from its source, how much melting occurred. Please don't include all of these possibilities, however it is important to simply insert errors based on calculation assumptions.

Reply. As suggested, we now briefly mention (lines 47-48) that the large variation in the estimates could partially be due to the assumptions of the calculations.

Lines 70-74. Please provide references for these statements. Further, there has been analyses of KREEPy magmatic lithologies for OH in apatite. Do these exhibit an abundant OH component?

Reply. The references below are cited in the revision as suggested. The OH-abundances of apatite from KREEPy lithologies overlap with those of CE5 basalt apatite as shown in Figure 2.

Jolliff B. L., Gillis J. J., Haskin L. A., Korotev R. L. and Wieczorek M. A. Major lunar crustal terranes: Surface expressions and crust-mantle origins. *Journal Geophysical Research* 105 (2000), 4197-4216.

Potts N. J., Bromiley G. D. and Brooker R. A. An experimental investigation of F, Cl and H₂O mineral-melt partitioning in a reduced, model lunar system. *Geochimica et Cosmochimica Acta* 294 (2021), 232-254.

Lines 152-154. This sentence does not adequately address the conclusion that little exchange of H between melt inclusions and ilmenite. Please expand this idea into an additional sentence.

Reply. The lowest δD values of the melt inclusions are -330 ± 190 ‰. Overlapping with the ranges of lunar mantle and chondritic water, these values are thus indicative of exceptional preservation of the H isotopes in the melt inclusions without significant H isotope exchange with the D-enhanced residual melt. The sentence has been revised accordingly (lines 141-147).

Figure 2. The addition of Apollo samples on this diagram subtracts from the observations made on the CE5 samples. I suggest that only the CE 5 data and interpretation is kept in this diagram. A comparison to the Apollo sample data is presented in a separate diagram in an Appendix.

Reply. After careful consideration and with due respect to the reviewer's opinion, we prefer to keep the Apollo data in the plot. Nonetheless, to honor the reviewer's reasonable wish to have more focus on the CE5 samples, we have modified the symbols to highlight the new data from CE5 basalts.

Lines 205-209. The model for producing an incompatible element enriched magma from a depleted source in the manuscript "Non-KREEP origin for Chang'E-5 basalt in the Procellarum KREEP Terrane" does not reproduce the REE pattern of the CE5 basalts. Therefore, the calculations here may have issues. I would suggest that the melting-crystallization model in Non-KREEP origin for Chang'E-5 basalt in the Procellarum KREEP Terrane is improved and then applied to this calculation. In this calculation, the liquid line of descent models by John Longhi could be used to confirm the modeled % crystallization after melt extraction.

Reply. The manuscript "Non-KREEP origin for Chang'E-5 basalt in the Procellarum KREEP Terrane" by our colleagues has been revised. In the revision, the results of their model being the low degrees (2-3%) of partial melting of the depleted mantle source and a following high degree (43-78%) of fractional crystallisation do not change, which can reproduce the REE patterns of the parent magma equilibrated with the core of clinopyroxene of CE5 basalts. But the bulk composition of the CE5 basalts contain even greater REE enrichment than their parent magma. Thus, in order to reproduce the bulk

basalt REE pattern, a higher degree of 78-88% (in comparison with 43-78%) of fractional crystallisation is required (Tian et al., submitted to Nature). Accordingly, we use the updated degree of 43-88% (instead of 43-78%) of fractional crystallisation to estimate the water abundance of the mantle source of CE5 basalts.

Tian, H.-C., Wang, H., Chen, Y., Yang, W., Zhou, Q., Zhang, C., Lin, H.-L., Huang, C., Wu, S.-T., Jia, L.-H., Xu, L., Zhang, D., Li, X.-G., Chang, R., Yang, Y.-H., Xie, L.-W., Zhang, D.-P., Zhang, G.-L., Yang, S.-H., Wu, F.-Y., 2021. A non-KREEP origin for the Chang'E-5 basalts in the Procellarum KREEP Terrane. Submitted to Nature.

Lines 231-233. This evolution of H content of the lunar mantle with time is not supported by the data in Figure 3. There appears to be numerous older basalts between 3.2 and 4.0 Ga that are also “depleted” in the H component of their sources. This appears not to be time dependent but mantle source dependent (e.g., LMO cumulate horizon, or previous melting). Perhaps, the authors should investigate if the low H component in the older lunar basalts is partially tied to the depleted nature of the source.

Reply. We agree that there is no obvious decreasing trend of water abundance in the lunar mantle with time, and our estimate of water abundance for the mantle source region of CE5 basalts overlaps with the low range of previous estimates. According to the suggestions, we explain the low water abundance of CE5 mantle source in two main scenarios, i.e. extensive dehydration by partial melting of mantle source via prolonged volcanic activity or the original water-poor nature of the CE5 mantle source.

Lines 241-248. It is an interesting premise that the H in the lunar mantle evolves with time. However, the data in Figure 3 does not support this model. The concept presented here is less exciting, but is more likely.

Reply. Addressed above.

Lines 256-257. Perhaps this is not much of an enigma as implied here. This part of the Moon has high concentrations of heat producing elements. It is possible that the high heat-flow associated with this region of the Moon enabled the melting of enriched and depleted mantle sources.

Reply. Because basaltic volcanism, according to our new CE5 age, lasted unusually late to 2 Ga, before this study, we expected high concentrations of heat-producing elements and/or high water abundance of the mantle source beneath the CE-5 landing site. However, the petrographic and geochemical features of CE5 basalt clasts have revealed that the mantle source region of CE5 basalts is depleted in heat-producing elements (Th and U) and other incompatible elements, not related with KREEP-components (Tian et al., 2021, submitted to Nature). Meanwhile, our analysis of water abundances and H isotopes in both apatite and ilmenite-hosted melt inclusions results in a water-poor mantle source for CE5 basalts too. So, we have not yet explained why this region has such young basalts, but we have critically eliminated the two previously preferred hypotheses.

The last paragraph (lines 208-219) has been re-written.

Tian, H.-C., Wang, H., Chen, Y., Yang, W., Zhou, Q., Zhang, C., Lin, H.-L., Huang, C., Wu, S.-T., Jia, L.-H., Xu, L., Zhang, D., Li, X.-G., Chang, R., Yang, Y.-H., Xie, L.-W., Zhang, D.-P., Zhang, G.-L., Yang, S.-H., Wu, F.-Y., 2021. A non-KREEP origin for the Chang'E-5 basalts in the Procellarum KREEP Terrane. Submitted to Nature.

Charles "Chip" Shearer

Response to Referee #2

Referee #2:

Comments to Nature manuscript 2021-07-12319 "A dry lunar mantle reservoir for young mare basalts of Chang'E-5" by Hu et al.

Hu et al. report new 'water' contents and D/H ratios for apatite and ilmenite-hosted melt inclusions in Chang'E-5 basaltic clasts determined by SIMS. Since the CE5 mission recently sampled young mare units, far from the Apollo and Luna landing sites, these results may provide crucial new constraints on the spatial distribution and temporal evolution of the water abundance in the Moon's interior. Notably, the new results suggest that the water concentration in the mantle source sampled by C5 basalts is very low.

While I am excited by the topic and analytical approach of this study, I have critical concerns about the 'water' contents and spallation-corrected δD values reported for the (low-H₂O) melt inclusions, as detailed below. Consequently, I am not convinced that the 3-stage evolutionary path proposed for the melt (which is mainly based on MI data) is reliable. Due to the concerns I have about the measurements and data treatment, I am unable to support publication of the manuscript in its current form.

Reply: According to the comments and suggestions, we have clarified the main concerns about the analyses of the low-water melt inclusions. In fact, as shown in Figure 2, our analyses of apatite and melt inclusions from CE5 basalts overlap with those of Apollo samples. However, CE5 samples were collected from the youngest basalt unit in the region, and they are arguably derived from one volcanic eruption. This simple context renders the new data clear and easy to explain.

1) SIMS analyses:

1.1) 12 out of the 23 studied clasts were prepared as thin sections mounted in epoxy. This is not ideal for H analyses by SIMS. The authors should clarify if H backgrounds (i.e., H signals of San Carlos olivine) and H signals of pyroxene were comparable for samples mounted in epoxy or Sn-Bi. This is important for assessing the accuracy of low-H₂O analyses.

Reply: We agree that epoxy may not be the ideal embedding material for H analysis, and only ~1/3 of the hand-picked clasts (~240 fragments with a longest dimension from ~100 μm to ~1 mm) were embedded in 3 epoxy mounts and prepared into polished thin sections (~100 μm in thickness), in an attempt to reduce the amount of epoxy. Most of the hand-picked clasts (~2/3 of the total) were made into 8 metal-alloy mounts. All of the sample mounts were baked overnight and stored in the NanoSIMS sample chamber to minimize the H background. The H background was monitored before and during analysis of the CE5 samples. We found that the $^1\text{H}/^{16}\text{O}$ ratios of CE5 silicates in epoxy mounts were even slightly lower than those in the metal alloy mounts, and both were comparable to the values measured on the anhydrous San Carlos olivine reference in the same sample holder (Fig. R1, see image below). These measurements demonstrate that the samples prepared with epoxy didn't increase the H background compared to the metal-alloy mounts.

Fig. R1. Monitoring of the H background. (a) Measured $^1\text{H}/^{16}\text{O}$ ratios and (b) calculated H_2O contents of silicates in the metal alloy and epoxy mounts of CE5 basalt clasts and San Carlos olivine (SCOL) in the same sample holder. The associated data sets have been compiled in Supplementary Table S2.

The vacuum quality in the analysis chamber of NanoSIMS 50L is the predominant factor affecting the H_2O background (Hu et al., 2015, Fig. R2). During the analysis of the CE5 samples, the vacuum in the analysis chamber was lower than $3\text{E}-10$ torr.

Fig. R2. The H (counts per second, CPS) signal intensities on both anhydrous sapphire and a silicon wafer are positively correlated with the vacuum of the analysis chamber (Figure 2 of Hu et al. (2015)). Note that axes are log scale.

Hu, S., Lin, Y.T., Zhang, J.C., Hao, J.L., Yang, W., Deng, L.W., 2015. Measurements of water content and D/H ratio in apatite and silicate glasses using a NanoSIMS 50L. *Journal of Analytical Atomic Spectrometry* 30, 967-978.

1.2) San Carlos olivine was used to determine the instrumental background of H.

The 'water' content of San Carlos is stated to be 1.4 ppm on Line 442. However, San Carlos olivine can contain 10's to 100's of ppm OH (see Kurosawa, M., Yurimoto, H. and Sueno, S. (1997), *Phys Chem Min* 24, 385), suggesting that it does not represent an appropriate dry standard for low-H₂O concentration measurements and/or instrumental background measurements. How can the authors be sure that their olivine grains contain only 1.4 ppm H₂O? This crucial issue must be addressed here.

Reply: SIMS analysis of water content for glass and anhydrous minerals, especially for samples/minerals with the low water abundance, was challenged by the instrumental water background before the work of Hauri et al. (2002). The more reliable water contents of San Carlos olivine were estimated to be ~1 ppm by Aubaud et al. (2007), which was widely used by the community thereafter. San Carlos olivine does have variation to some extent in water content, but it is likely induced by nanometer-sized pores (Mosenfelder et al., 2011). Zhang et al. (2019) analyzed the water contents of San Carlos and other olivine references with FTIR, and obtained a calibration line for SIMS analysis for the water content of olivine (Fig. R3). They reported 1.4 ppm of water for San Carlos olivine. We have analyzed the same San Carlos olivine grains (Zhang et al., 2019) before this study, and our results of the grains used in our lab are comparable to those of previous work.

Because the water content of San Carlos olivine (~1.4 ppm) is much lower than the instrumental background (25 ± 8 ppm), the counts measured on San Carlos olivine were referred to as the H₂O background. The very low water content of San Carlos olivine is also confirmed by the analyses on the nominally anhydrous silicates of CE5 basalt clasts, where the two results overlap with each other (Fig. R1). Even if the olivine grains contain a few ppm of water (less than the uncertainty of the background), our analyses could be

underestimated by a few ppm of water, but this would still be within the analytical uncertainty.

Fig. R3. Water content calibration line established on olivine standards (San Carlos (SCOL), KLB-1, ICH-30, and Mongok) without H₂O background subtraction (Zhang et al., 2019).

Aubaud, C., Withers, A.C., Hirschmann, M.M., Guan, Y., Leshin, L.A., Mackwell, S.J., Bell, D.R., 2007. Intercalibration of FTIR and SIMS for hydrogen measurements in glasses and nominally anhydrous minerals. *American Mineralogist* 92, 811.

Hauri, E., Wang, J.H., Dixon, J.E., King, P.L., Mandeville, C., Newman, S., 2002. SIMS analysis of volatiles in silicate glasses 1. Calibration, matrix effects and comparisons with FTIR. *Chemical Geology* 183, 99-114.

Mosenfelder, J.L., Le Voyer, M., Rossman, G.R., Guan, Y., Bell, D.R., Asimow, P.D., Eiler, J.M., 2011. Analysis of hydrogen in olivine by SIMS: Evaluation of standards and protocol. *American Mineralogist* 96, 1725-1741.

Zhang, W.F., Xia, X.P., Eiichi, T., Li, L., Yang, Q., Zhang, Y.Q., Yang, Y.N., Liu, M.L., Lai, C., 2019. Optimization of SIMS analytical parameters for water content measurement of olivine. *Surface and Interface Analysis* 52, 224-233.

The authors state that the H₂O background determined from SC analyses is 25 ± 8 ppm. Nonetheless, it appears that the water content calibration line (Extended Data Fig. 2) was forced through zero. This suggests that the background correction is wrong. This would

greatly affect the calculated water contents of low-H₂O samples (i.e., melt inclusions). Please clarify.

Reply: All analyses were first subtracted for the H₂O background determined with the nominally anhydrous San Carlos olivine reference. The background was subtracted following the correction method of Tartèse et al. (2019). This is important for the low-H₂O melt inclusions.

After subtraction of the H background, the water contents of samples were calculated from the background-subtracted ¹H/¹⁶O ratio multiplied by the slope of the calibration line. Because the H₂O background is very low, the slope of the calibration line is almost the same if it is forced through the background (25 ± 8 ppm) or zero.

In Extended Data Figure 2, it is noted that 2 apatite standards, 3 basaltic glass standards and the San Carlos olivine reference all plot on a straight line, i.e. the calibration line, which is suggestive of similar matrix effects for ¹H/¹⁶O⁻ of both apatite and silicate glass. The matrix effects of apatite and silicate glass are also commented on below in detail to another related response.

Tartèse, R., Anand, M., Franchi, I.A., 2019. H and Cl isotope characteristics of indigenous and late hydrothermal fluids on the differentiated asteroidal parent body of Grave Nunataks 06128. *Geochimica Et Cosmochimica Acta* 266, 529-543.

Also note that on Line 103 of the suppl. info, the reported instrument background of H is 5 ± 2 ppm. This is confusing.

Reply: The H₂O background is negatively correlated with the Cs⁺ beam current. We used a higher beam current (7 nA) to analyse the nominally anhydrous pyroxene, which significantly reduces the H₂O background to ~5 ppm. This confusion has been clarified in the revision.

1.3) Extended Data Fig. 2 suggests that one single water content calibration line was used that includes both apatite and glass standards. This is unusual. Does this suggest that there is not matrix effect for glasses and apatite (and cpx)? Again, this may be important for low-H₂O analyses.

Reply: We have tested three analytical modes of determining the matrix effect between apatite and silicate glasses over a period of 3 years. (1) Peak jump-multicollection combined isotope mode: secondary ions ¹H⁻ and ²D⁻ under magnetic field-1 (B1), ¹²C⁻ and ¹⁸O⁻ under magnetic field-2 (B2), cycling magnetic field for an analysis. (2) Multicollection isotope mode: secondary ions ¹H⁻, ²D⁻, ¹²C⁻ and ¹⁸O⁻ were collected simultaneously under the same magnetic field. (3) Multicollection element mode: ¹²C⁻, ¹⁶O¹H⁻, ¹⁸O⁻, ³²S⁻ and ³⁵Cl⁻ were collected simultaneously. In the two isotope modes (Fig. R4a and R4b), with ¹H/¹⁸O used as the water content index, the matrix effects for apatite and silicate glasses are identical within analytical uncertainties when samples contain <~ 2wt% in water content (Hu et al., 2015). In contrast, in the element mode, with ¹⁶O¹H/¹⁸O used as the water content index, the calibration line of apatite differs markedly from that of silicate glasses

(slopes: 3.7 vs. 0.87, Fig. R4c) (Hu et al., 2015). Furthermore, the similar matrix effects for apatite and silicate glasses were also replicated by different laboratories: the NanoSIMS labs at the Open University (Tartèse et al., 2019) and at the Carnegie Institute of Washington (Sarafian et al., 2019) under the isotope mode (^1H , ^2D , ^{12}C , and ^{18}O). Thus, in this study, the apatite and silicate glass standards are plotted on the same straight line in the water content vs $^1\text{H}/^{16}\text{O}$ figure (Extended Data Fig. 2).

As mentioned above, for the low- H_2O silicate melt inclusions, it is important to subtract the H_2O background first. The water contents of the melt inclusions were then calculated from the background-subtracted $^1\text{H}/^{16}\text{O}$ ratios and the slope of the calibration line. We do not attempt to quantitatively determine the water content of CE5 clinopyroxene, because the value could be even lower than the uncertainty of the H_2O background.

In summary, the matrix effect issue has been clarified in the revision.

Fig. R4. The matrix effects of three analytical modes for water content calibration (Hu et al., 2015).

Hu, S., Lin, Y.T., Zhang, J.C., Hao, J.L., Yang, W., Deng, L.W., 2015. Measurements of water content and D/H ratio in apatite and silicate glasses using a NanoSIMS 50L. *Journal of Analytical Atomic Spectrometry* 30, 967-978.

Sarafian, A.R., Nielsen, S.G., Marschall, H.R., Gaetani, G.A., Righter, K., Berger, E.L., 2019. The water and fluorine content of 4 Vesta. *Geochimica Et Cosmochimica Acta* 266, 568-581.

Tartèse, R., Anand, M., Franchi, I.A., 2019. H and Cl isotope characteristics of indigenous and late hydrothermal fluids on the differentiated asteroidal parent body of Grave Nunataks 06128. *Geochimica Et Cosmochimica Acta* 266, 529-543.

1.4) Extended Data Fig. 3 indicates that the measured δD values of the standard glass and apatite were "consistent with their recommended values". The lack of any instrumental mass fractionation is highly unusual. Does this imply that no correction of IMF was applied?

Reply: (1) Similar to the calibration for water contents of apatite and silicate glasses, the IMF of apatite and silicate glasses are identical within analytical uncertainties ($2SD < 45\%$) (Fig. R5) (Figure 5 of Hu et al., 2015). (2) In this study, IMF was corrected using the Kovdor apatite standard during the whole analytical session for apatite and melt inclusion glasses of CE5 basalt clasts. (3) Meanwhile, the standards SWIFT MORB glass and Durango apatite were also measured as references, and their calibrated δD values are consistent with the recommended values within their respective analytical uncertainties (Extended Data Figure 3). In addition, the calibrated δD values of the basaltic glasses 1833-11 and 519-4-1 are also consistent with the values of Earth's mantle within errors (i.e., Hallis et al., 2015, *Science*).

Thus, the IMF issue has been clarified in the revision.

Fig. R5. IMF of D/H in apatite and MORB glass (Hu et al., 2015).

Hallis, L.J., Huss, G.R., Nagashima, K., Taylor, G.J., Halldórsson, S.A., Hilton, D.R., Mottl, M.J., Meech, K.J., 2015. Evidence for primordial water in Earth's deep mantle. *Science* 350, 795-797.

Hu, S., Lin, Y.T., Zhang, J.C., Hao, J.L., Yang, W., Deng, L.W., 2015. Measurements of water content and D/H ratio in apatite and silicate glasses using a NanoSIMS 50L. *Journal of Analytical Atomic Spectrometry* 30, 967-978.

1.5) Fig. 1b and Fig. S1 indicate that many SIMS spots in melt inclusions included mineral phases (as opposed to a glass phase only). This is also confirmed by the EMPA results reported in Table S2. This implies that a correction needs to be applied to their H₂O content since the remaining melt will be enriched in H compared to the crystallized minerals. Notably, the inclusion that is considered to be the most primitive (103-020,013) appears to be entirely re-crystallized. This important issue must be addressed and discussed.

Reply. We agree that the post-entrapment crystallisation (up to 52 %) of the melt inclusions will enhance the water contents in the remaining glass. According to the comments, we have made a correction for the post-entrapment crystallisation effects on the melt inclusions, and the original and corrected values have been compiled in Extended Data Table 3 and Supplementary Table S4.

Could the high water content of MI 406-015,045 be explained by a contribution of 'water' in nearby fractures?

Reply. We do not think the high water abundance (661 ppm, the original analysis) of MI 406-015,045 was due to the possible contribution of "water" from nearby fractures. This melt inclusion plots close to the low D/H range of apatite (Fig. 2), and can be adequately explained by the capturing of the final residual melt when apatite started to precipitate.

2) Correction of D/H ratios for spallation effects

The D production rate from Merlivat et al. (1976) was used to correct the D/H ratios of apatite. Why? This value was established for a basalt. Hashizume et al. (EPSL, 2002) and Fűri et al. (GPL, 2018) showed that single grains within a regolith sample have highly variable exposure ages that can range up to 400 Ma (even up to >1Ga). Therefore, the use of a nominal CRE age of 50 Ma for all CE5 basalt clasts is unlikely to be correct. This suggests that the calculated D/H ratios of low-H₂O samples (i.e., melt inclusions) are wrong. Knowledge of the CRE age of each individual grain is necessary for an accurate spallation correction.

Reply. The correction of D/H ratios for spallation is dependent on D production rate and CRE age.

(1) For melt inclusions, we used the D production rate re-evaluated by Fűri et al. (2017); for apatite, we found only the data reported by Merlivat et al. (1976). However, the spallation correction of D/H ratios for apatite can be ignored, because the majority of apatite grains contain >500 ppm water.

(2) We agree that individual clasts have different CRE ages, and the clasts studied here have not been determined for CRE ages. In this case, we made calculations using different CRE ages, i.e. 10, 50, 100, and 200 Ma. The calculations show that the results

will be over-corrected if CRE ages of 100 and 200 Ma are used. **The response has been redacted as it relates to unpublished material.**

(3) As suggested, in the revision we now mention that some grains of Apollo samples have CRE ages up to 400 Ma (Füri et al, 2017).

(4) Even if our clasts have older CRE ages, and the D/H ratios of the melt inclusions are lower to some extent than the corrected values, this nonetheless does not change one of the main discoveries that the parent magma of CE5 basalts was D-depleted. Accordingly, irrespective of the actual CRE ages, the highly enhanced D/H ratios of apatite grains reflect D-enrichment via H₂ degassing of the magma.

Füri, E., Deloule, E., Trappitsch, R., 2017. The production rate of cosmogenic deuterium at the Moon's surface. *Earth and Planetary Science Letters* 474, 76-82.

Merlivat, L., Leiu, M., Neif, G., Roth, E., 1976. Spallation deuterium in rock 70215, Lunar and Planetary Science Conference Proceedings, pp. 649-658.

3) Melt inclusions in ilmenite - Water abundance of the CE5 mantle source

3.1) Three melt inclusions were analyzed twice (see Extended Data Table 3), and the measured water abundances and/or D/H ratios are highly variable. This observation should be discussed and explained.

Reply. For the melt inclusions with a relatively large size, we attempted to repeat the analysis. Two of the melt inclusions (clast No. 103-020,013 and 103-020,018) show variations in water abundances larger than analytical uncertainty. This variation could be attributed to partial covering on tiny daughter silicates crystallized in melt inclusions. We have clarified this issue in the revision.

The D/H ratios of these three melt inclusions are reproducible within the analytical uncertainty.

3.2) "the water-richest melt inclusions have δD values of $\sim -300\%$ " (Supp. Info. Line 62) - this is not true! The water-richest MI has a δD value of $+271\%$. This misleading statement must be rephrased. (Please also rephrase the sentences on Lines 177 and 199-200 of the main text.)

Reply. As suggested, we have revised the statement in question to now read: "The water-rich and D-poor end-member of these melt inclusions have δD values of $-330 \pm 190 \%$ " (Lines 534-535).

The misleading sentences in question have also been revised.

3.3) Which water partition coefficient was used to derive the water content of the mantle source? Is this value well known for lunar mantle conditions? Please report these details in section "5.1." (should be 2.1?) of the suppl. info.

Reply. The partition coefficient of water between the derived basaltic melt and the mantle source materials is ≤ 0.03 (Potts et al., 2021). We simply assume all water was extracted from the mantle source by the melt.

We have integrated these parts into the Methods and re-organized as "Estimate of water abundance for the mantle source of CE5 basalts" (Lines 578-588) according to the author guidelines.

Potts, N.J., Bromiley, G.D., Brooker, R.A., 2021. An experimental investigation of F, Cl and H₂O mineral-melt partitioning in a reduced, model lunar system. *Geochimica Et Cosmochimica Acta* 294, 232-254.

4) δD value of the CE5 mantle source

The parent magma (and mantle source) of CE5 basalts is estimated to have a δD value of $\sim -330\text{‰}$. Please briefly state what this value could indicate about the origin of water on the Moon.

Reply. We briefly mentioned that the δD value of $\sim -330 \pm 190 \text{‰}$ is within the ranges of the lunar mantle and chondrites. But a discussion of the wide range of δD (~ -200 to $+200\text{‰}$) of the lunar interior is beyond the scope of this contribution.

The CE5 mantle source is expected to have experienced multiple water-bearing melt extraction episodes (Lines 234-235). Are these melt extraction events expected to have modified the initial δD value of the mantle source?

Reply. Partial melting of the terrestrial and lunar mantle materials does not fractionate D/H significantly (Kyser, 2018).

Based on samples collected from different petrographic locations and formed from 4 to 2 Ga, the δD value estimates for the lunar mantle are within a range of ~ -200 to $+200\text{‰}$. In comparison with the large D-enrichment of apatite from the same basalts, the lunar mantle δD range is relatively small, suggestive of no significant isotope fractionation for the mantle source via partial melting processes.

Kyser, T.K., 2018. Stable Isotopes in High Temperature Geological Processes, in: John, W.V., Hugh, P.T., James, R.O.N. (Eds.), Chapter 5. STABLE ISOTOPE VARIATIONS in the MANTLE. De Gruyter, pp. 141-164.

Additional comments/questions:

Line 86: apatite is "F-rich and Cl-poor"?

Reply. Yes, revised as suggested.

Line 382: I presume the sections "were initially coated with carbon" for SEM analyses and then re-coated with Au. Please clarify.

Reply. We coated the samples with Au first for both SEM and NanoSIMS analysis. After NanoSIMS analysis, they were re-coated with C and observed with SEM to confirm the NanoSIMS analysis spots, followed by EPMA analysis. The sample coating procedure has been clarified in the revision.

Lines 412-413: It is not clear which ions were measured on EMs or a Faraday cup. Please rephrase.

Reply. As suggested, the sentence has been revised to read: "the secondary anions $^1\text{H}^-$, $^2\text{D}^-$, and $^{12}\text{C}^-$ were simultaneously counted by electron multipliers (EMs) and $^{16}\text{O}^-$ by a Faraday cup".

Line 420: Please give the units (cps?) for Hbg.

Reply. Yes, revised as suggested.

Line 429: "more details can be found in Hu et al." Extended Data Tables 2 and 3: The uncertainties of the δD values for CRE age = 50 Ma should be reported because these values are used in the discussion.

Reply. 50 ‰ (2σ) of the uncertainty on δD values raised by spallation correction is used, following Saal et al. (2013). This has been clarified in the revision and the associated values have been updated in the main text and data sets.

Saal, A.E., Hauri, E.H., Van Orman, J.A., Rutherford, M.J., 2013. Hydrogen isotopes in lunar volcanic glasses and melt inclusions reveal a carbonaceous chondrite heritage. *Science* 340, 1317-1320.

Extended Data Fig. 1, Line 580: "represented by a high Th abundance"? Please rephrase/clarify.

Reply. Revised to read: "Chang'E-5 landed within the northwestern Oceanus Procellarum KREEP-rich terrain (outlined) in a region with a high Th abundance".

Extended Data Fig. 3: $\delta\text{D}_{\text{SMOW}}$ - correct y-axis label

Reply: Revised as suggested. The symbols are also updated in accordance with Figure 2.

Supp. Info. Line 61: "supply with convincing evidence" - rephrase.

Reply: Revised to read: "The negative correlation of the melt inclusions can be explained by degassing of H_2 in the basaltic magma and D-enrichment during this process".

Supp. Info. Line 67: "pyroxene and olivine could have deposited" - rephrase.

Reply. Revised to read: "pyroxene and olivine could have crystallised".

Supp. Info. Line 78: "regardless large variation" - not clear, please rephrase.

Reply. These sentences have been deleted.

Supp. Info. Line 100: "has be investigated" - replace by "has been found to be"

Reply. This paragraph has been rewritten.

Table S1: Please explain what you mean by "size" and "total area".

(Maybe it would make more sense to report the areas in μm^2 .)

Reply. "size" is replaced by "dimension" and "total area" has been revised as "area"

Table S2: The TiO_2 content of ilmenite is not correctly reported.

Reply: Reporting of the TiO_2 contents has been corrected.

Table S3: Please check the format of the reported data (e.g. Err Mean of the 2H/1H ratio).

Reply: All of the reported data sets have been carefully checked and updated accordingly.

Which results are shown in the column labeled "Ratio"?

Reply: It is $^{12}\text{C}/^{16}\text{O}$ ratio, and has been revised as such.

*****END*****

Reviewer Reports on the First Revision:

Referee #1:

I reviewed this document a second time. It is my impression that the authors responded to my initial review in an appropriate manner. I don't think that I have access to the second reviewers comments to view if the authors were responsive to their comments.

Referee #2:

Comments to Nature manuscript 2021-07-12319A "A dry lunar mantle reservoir for young mare basalts of Chang'E-5" by Hu et al.

The marked-up version of the revised manuscript and the responses clearly show that the authors have considered and addressed all my comments. I still think that the uncertainty of spallation-corrected dD values is more significant than acknowledged here (for the MIs), but I agree that this uncertainty has "only minor effects on estimating the maximum water abundances for CE5 parent magma and the mantle source". I have no additional comments, and I highly recommend the paper for publication in Nature.

A few minor typing mistakes will have to be corrected:

Line 35: "is at the low end of the range estimated..."

Lines 126-127: as determined from Apollo -> not clear

Line 222: "and is water-poor"

Line 409: I think "torr" should be "Torr"

Line 466: [values] "overlap"

Line 467: low-water melt "inclusions"

Line 488: "grains"

Line 563-567: The sentence seems incomplete.

Line 579: "under reduced lunar conditions"

Line 699: "Terrane" instead of "terrain" - see main text

Lines 730 and 734: "Uncertainties include"

Line 734: remove "will"

Author Rebuttals to First Revision:

Referee #2:

Comments to Nature manuscript 2021-07-12319A "A dry lunar mantle reservoir for young mare basalts of Chang'E-5" by Hu et al.

The marked-up version of the revised manuscript and the responses clearly show that the authors have considered and addressed all my comments. I still think that the uncertainty of spallation-corrected dD values is more significant than acknowledged here (for the MIs), but I agree that this uncertainty has "only minor effects on estimating the maximum water abundances for CE5 parent magma and the mantle source". I have no additional comments, and I highly recommend the paper for publication in Nature.

A few minor typing mistakes will have to be corrected:

Line 35: "is at the low end of the range estimated..."

Reply: revised as suggested (Line 35).

Lines 126-127: as determined from Apollo -> not clear

Reply: revised as "As determined from Apollo samples" (Line 128)

Line 222: "and is water-poor"

Reply: revised as suggested (Line 224).

Line 409: I think "torr" should be "Torr"

Reply: revised as suggested (Line 419).

Line 466: [values] "overlap"

Reply: revised as suggested (Line 476).

Line 467: low-water melt "inclusions"

Reply: revised as suggested (Line 477).

Line 488: "grains"

Reply: revised as suggested (Line 497).

Line 563-567: The sentence seems incomplete.

Reply: This long sentence has been re-organized as below:

As discussed above, the parent magma of CE5 basalts has the original δD value of ~ -330 ‰ indicated by the most D-depleted melt inclusions, **whereas** the D-enrichment of apatite was **likely** attributed to degassing of water in the form of H_2 . **Hence**, the water abundance of the parent magma was calibrated to be $600 \pm 400 \mu g \cdot g^{-1}$, with 98-99% water degassing loss

required to enhance δD values from $\sim -330\text{‰}$ to $\sim 600\text{‰}$ based on the H_2 degassing modeling.
(Lines 572-576)

Line 579: "under reduced lunar conditions"

Reply: revised as suggested (Line 588).

Line 699: "Terrane" instead of "terrain" - see main text

Reply: revised as suggested (Line 685).

Lines 730 and 734: "Uncertainties include"

Reply: revised as suggested (Line 720).

Line 734: remove "willl"

Reply: revised as suggested.